

**Quantification of multiple simultaneously occurring**
**nitrogen flows in the euphotic ocean**
Min Nina Xu[1], Yanhua Wu[2], Li Wei Zheng[1], Zhenzhen Zheng[1], Huade Zhao[3],
Edward A Laws[4], Shuh-Ji Kao[1]*
[1]State Key Laboratory of Marine Environmental Science, Xiamen University, Xiamen,
China
[2]Shenzhen Marine Environment Monitoring Center Station, Shenzhen, China
[3]National Marine Environmental Monitoring Center, Dalian, China
[4]School of the Coast & Environment, Louisiana State University, USA
*Correspondence to*: Shuh-Ji Kao (sjkao@xmu.edu.cn)



**Abstract**
The general features of the N cycle in the sunlit ocean are known, but quantitative
information about multiple transformation rates among nitrogen pools, i.e., ammonium
($NH_4^+$), nitrite ($NO_2^-$), nitrate ($NO_3^-$) and particulate/dissolved organic nitrogen
(PN/DON), are limited due to methodological difficulties. By adding a single
$^{15}N$-labelled $NH_4^+$ tracer into incubators, we monitored the changes in concentration
and isotopic composition of the total dissolved nitrogen (TDN), PN, $NH_4^+$, $NO_2^-$, and
$NO_3^-$ pools to trace the $^{15}N$ and $^{14}N$ flows. Based on mass conservation and isotope
mass balance, we formulated a matrix equation that allowed us to simultaneously
derive the rates of multiple transformation processes in the nitrogen reaction web. We
abandoned inhibitors and minimized the alteration of the system by adding a limited
amount of tracer. In one single incubation, solution of the matrix equation provided the
rates of $NH_4^+$, $NO_2^-$, and $NO_3^-$ uptake; ammonia oxidation; nitrite oxidation; nitrite
excretion; DON release; and potentially, the remineralization rate. To our knowledge,
this is the first and most convenient method designed to quantitatively and
simultaneously resolve complicated nitrogen transformation rates, albeit with some
uncertainties. Field examples are given, and comparisons with conventional labeling
methods are discussed.
**Keywords**
Ammonium oxidation, isotope matrix, new production, nitrification, regenerated
production



## 1. Introduction

Nitrogen (N), which is an essential element in organisms' metabolic processes,
regulates productivity as a limiting nutrient in the surface waters of many parts of the
ocean (Falkowski, 1997; Zehr and Kudela, 2011; Casciotti, 2016). In the euphotic zone,
nitrogen rapidly interconverts among five major N compartments: particulate organic
nitrogen (PN), dissolved organic nitrogen (DON), ammonium ($NH_4^+$), nitrite ($NO_2^-$),
and nitrate ($NO_3^-$) (Fig. 1 a). Quantitative information on transformation rates in the
marine N-cycle may advance our understanding of the coupling of autotrophic and
heterotrophic processes involving carbon and nitrogen and the efficiency of the
biological pump. Such information would also facilitate evaluation of ecosystem
functions. However, the dynamic nature and complexity of the reactions involving
nitrogen make it a difficult task to resolve the rates of multiple simultaneous nitrogen
transformations. Inventory and isotope tracer methods have often been used in previous
studies; however, determining the rate of a specific process is difficult without using
inhibitors to block other processes that affect the concentrations of products and
reactants, and of course rates of nitrogen transformations are affected by a variety of
abiotic conditions (e.g., light and dark; Ward, 2008, 2011 and references therein).
The inventory method is often used to determine the uptake rates of ammonium, nitrite,
nitrate, and urea (McCarthy and Eppley, 1972; Harvey and Caperon, 1976; Harrison
and Davis, 1977; Dugdale and Wilkerson, 1986; Howard et al., 2007) and to examine
the occurrence and rate of nitrification (Wada and Hatton, 1971; Pakulski et al., 1995;



Ward, 2011 and references therein). However, because the concentrations of all forms
of nitrogen are affected by multiple processes, inhibitors have typically been added to
isolate the effects of specific processes. For example, the concentration of ammonium
is simultaneously manily controlled by phytoplankton consumption (PN as the product),
nitrifier utilization (nitrite/nitrate as the product), and addition via remineralization
from heterotrophic bacterial metabolism, zooplankton excretion, and viral lysis.
Therefore, inhibitors (parallel experiments with various inhibitors) have been applied
in many studies to block confounding processes (Bianchi et al., 1997; Santoro et al.,
2010a; Newell et al., 2011; Ward, 2011; Fernandez and Farías, 2012; Grundle and
Juniper, 2011; Grundle et al., 2013; Martens-Habbena et al., 2015). Unfortunately, the
addition of inhibitors may cause undesirable side effects (Ward, 2011; Ward, 2008 and
references therein).
The $^{15}$N-labeled tracer technique has been widely used as a direct measure of specific
nitrogen processes since the emergence of isotope ratio mass spectrometry (IRMS). For
example, the addition of $^{15}$N-labeled nitrate has been applied to estimate new
production (Dugdale and Goering, 1967; Chen, 2005; Painter et al., 2014). Likewise, by
incubating water to which $^{15}NH_4^+$ has been added under dark and light conditions, rates
of nitrification ($^{15}NO_3^-$ as product) have been measured (e.g. Newell et al., 2013; Hsiao
et al., 2014; Peng et al., 2016) and rates of ammonium uptake (regenerated production)
($^{15}N_{PN}$) (e.g. Dugdale and Goering, 1967; Dugdale and Wilkerson, 1986; Bronk et al.,
1994, 2014). However, ammonium uptake, nitrification, nitrate uptake, and ammonium



excretion occur simultaneously not only in the incubation bottles but also in the field.
For instance, Yool et al. (2007) synthesized available global data and indicated that the
fractional contribution of nitrate derived from nitrification in the euphotic zone to
nitrate uptake can be as high as 25–30%. Unfortunately, nitrate uptake rates were
determined under light conditions, and nitrification was determined under dark
conditions (Grundle and Juniper, 2011; Grundle et al., 2013), which are not comparable
in terms of their effects on these processes. To overcome this problem, 24-h incubation
have been used to compensate for the diel cycle of light-sensitive processes (Beman et
al., 2012). However, 24-h incubations may cause calculation artifacts due to the
interference from significant, multiple transfers of $^{15}$N and $^{14}$N among pools. An
innovative method to simultaneously measure multiple N flows is therefore needed to
more realistically resolve nitrogen transformations in the euphotic zone. Marchant et al.
(2016) have reviewed recent advances in marine N-cycle studies using $^{15}$N-labeling
substrates combined with nanoSIMS, FISH, or HISH. These methods provide
information about the N cycle at the cellular and molecular level. Nevertheless, the
rates of multiple transfers among compartments have not yet been measured
simultaneously within a community of microorganisms.
In this study, we propose an "isotope matrix method" that is simple in concept. To
avoid perturbations, the concentration of the tracer was limited to < 20 % of the
substrate concentration, as suggested by previous researchers (Raimbault and Garcia,
2008; Middelburg and Nieuwenhuize, 2000; Painter et al., 2014). One single tracer,



$^{15}NH_4^+$, was added to trace the $^{15}N$ flows among the nitrogen pools under simulated *in*
*situ* conditions. Almost all well accepted important processes in the N cycle can be
quantified with this newly proposed method. To demonstrate the applicability of the
method, we conducted incubation experiments in high-nutrient, coastal water off
southeastern China and in low-nutrient water in the western North Pacific. We found
that the success of the method was determined by the analytical precision of the
isotopic ratios of the dissolved pools, especially when concentrations were low.
Application of the method was facilitated by advances in the analytical methods used to
determine the concentration and isotopic composition of various nitrogen species. Use
of the new method allowed us to realistically quantify surface water nitrogen dynamics.
The method was also validated using the STELLA model.
**2.  Isotope matrix method**
**2.1 Framework of the inter-connections among nitrogen pools**
In the oxygenated and well-lit euphotic zone, the transformations of N between $NH_4^+$,
$NO_2^-$, $NO_3^-$, PN, and DON, are shown in Fig. 1. PN is operationally defined as the
organic nitrogen of particles trapped on a GF/F filter. Dissolved inorganic nitrogen
(DIN) and DON are the inorganic and organic nitrogen, respectively, in the dissolved
fraction that passes through a polycarbonate membrane with a 0.22 $\mu$m pore size. Since
DON includes the N in numerous dissolved organic N compounds, including
unidentified organics, urea, amino acids, amines, and amides, DON represents the



"bulk" DON and is calculated by subtracting the concentrations of $NH_4^+$, $NO_2^-$, and
$NO_3^-$ (DIN) from the total dissolved N (TDN).
As illustrated by the Michaelis-Menten equation (MacIsaac and Dugdale, 1969), a
zero-order reaction is a reaction for which the rate F is constant and independent of the
substrate (nitrogen herein) concentration. In other words, the reaction kinetics are
substrate-saturated, even when the substrate concentration decreases with time. In
contrast, in the nitrogen-poor oligotrophic ocean, the substrate concentration is
relatively low, and the reaction is a first-order reaction. In that case, the reaction rate is
directly proportional to the substrate concentration; however, the specific rate $(k, h^{-1})$ is
constant. We thus consider two different types of schemes in our method: high nitrogen
and low nitrogen (Fig. 1 a and b). Here, we describe the high nitrogen case as an
example.
The consumption of reactive inorganic nitrogen ($NH_4^+$, $NO_2^-$, and $NO_3^-$) is dominated
by photosynthetic uptake by phytoplankton ($F_1$, $F_3$, and $F_5$ in Fig. 1a). Due to DIN
assimilation by phytoplankton, the PN pool may increase, but DON ($F_8$ in Fig. 1a)
(Bronk and Glibert, 1993, Bronk et al., 1994; Bronk and Ward, 2000; Varela et al.,
2005) and $NO_2^-$ ($F_7$ in Fig. 1a) may be released (Wada and Hatton, 1971; Collos, 1998;
Flynn and Flynn, 1998; Lomas and Lipschultz, 2006) during assimilation. Besides
being reduced by phytoplankton uptake, the concentration of $NH_4^+$ may be increased by
remineralization ($F_2$ in Fig. 1a) and reduced by nitrification. Nitrification consists of





two basic steps: ammonia oxidation by archaea/bacteria (AOA/AOB) that oxidize
ammonia to nitrite ($F_4$ in Fig. 1a) and nitrite oxidation to nitrate by nitrite-oxidizing
bacteria (NOB) ($F_6$ in Fig. 1a). Note that recent studies have revealed a single
microorganism that may completely oxidize $NH_4^+$ to $NO_3^-$ (comammox) (Daims et al.,
2015; van Kessel et al., 2015), but its importance in the marine environment remains
unclear. Specific mechanisms or processes such as grazing and viral lysis may also
change the concentrations of $NH_4^+$, nitrite, and DON. However, the scope of this study
is to determine the nitrogen flows and exchanges among the often measured and
operationally defined pools of nitrogen. In this context, grazers and viruses belong to
the operationally defined PN and DON pools, respectively. Thus the roles of processes
such as grazing and viral lysis are incorporated in the paradigm depicted in Fig. 1.
The paradigm for the low nutrient case (Fig. 1b) is the same as the paradigm for the
high nutrient case, except that we combined $NO_2^-$ and $NO_3^-$ into $NO_x^-$.
**2.2 Analytical methods to determine the amounts of $^{15}N/^{14}N$ in various pools**
Our newly proposed method basically couples the $^{15}N$-labelling and inventory methods.
To trace the $^{15}N$ movement among pools, changes in the concentration and isotopic
composition of the target pools need to be determined. Analytical methods to determine
the concentrations and isotopic compositions of both high and low levels of
inorganic/organic nitrogen are in most cases well established and have been reported



elsewhere. We determined all of the mentioned concentrations and isotopic
compositions except the isotopic composition of $NH_4^+$.
Concentrations of $NH_4^+$ higher than 0.5 $\mu$M were measured manually by using the
colorimetric phenol hypochlorite technique (Koroleff, 1983). Nanomolar $NH_4^+$
concentrations were measured by using the fluorometric $o$-phthaldialdehyde (OPA)
method (Zhu et al., 2013). Concentrations of $NO_2^-$ and $NO_x^-$ ($NO_2^-$ + $NO_3^-$) were
determined with the chemiluminescence method following the protocol of Braman and
Hendrix (1989). The detection limits of $NO_2^-$ and $NO_x^-$ were both ~ 10 nmol $L^{-1}$, and
the corresponding relative precision was better than 5% within the range of
concentrations that we measured. By using persulfate as an oxidizing reagent, we
oxidized TDN and PN separately to nitrate (Knapp et al., 2005) and then measured the
nitrate by using the analytical method for $NO_x^-$ described above.
We determined the $\delta^{15}N$ of $NO_2^-$ with the azide method by following the detailed
procedures in McIlvin and Altabet (2005). The $\delta^{15}N$ of $NO_x^-$ was determined by using a
distinct strain of bacteria that lacked $N_2O$ reductase activity to quantitatively convert
$NO_x^-$ to nitrous oxide ($N_2O$), which we then analyzed by IRMS (denitrifier method;
(Sigman et al., 2001; Casciotti et al., 2002). The isotopic composition of $NO_3^-$ was
determined from isotope mass balance ($NO_x^-$ minus $NO_2^-$) or measured by the
denitrifier method after eliminating preexisting $NO_2^-$ with sulfamic acid (Granger and
Sigman, 2009). To determine the $\delta^{15}N$ of TDN and PN, both species were converted to



$NO_3^-$ with the denitrifier method and the $\delta^{15}N$ of the $NO_3^-$ was determined as described
above. The most popular way to determine the N isotopic composition of $NH_4^+$ is the
"diffusion method", which involves conversion of dissolved $NH_4^+$ to $NH_3$ gas by
raising the sample pH to above 9 with magnesium oxide (MgO) and subsequently
trapping the gas quantitatively as $(NH_4)_2SO_4$ on a glass fiber (GF) filter; the isotope
ratios of the $^{15}N/^{14}N$ are then measured using a coupled elemental analyzer with an
IRMS (Holmes et al., 1998; Hannon and Böhlke, 2008). Alternatively, after removing
the preexisting $NO_2^-$ from the seawater samples using sulfanilic acid, $NH_4^+$ is first
quantitatively oxidized to $NO_2^-$ by hypobromite $(BrO^-)$ at pH ~12 $(BrO^-$ oxidation
method), and the protocol of McIlvin and Altabet (2005) is then used to reduce the
$NO_2^-$ to $N_2O$ (Zhang et al., 2007). Unfortunately, neither of these methods has been
established in our lab yet. The isotope matrix method requires the isotopic composition
of $NH_4^+$ as well, but this requirement can be circumvented by making certain
assumptions, as illustrated in our case studies.
We estimated the amount of $^{14}N$ and $^{15}N$ atoms in every individual pool for which we
knew the concentration and $\delta^{15}N$. By assuming the $^{15}N$ content of standard atmospheric
nitrogen to be 0.365%, we used the $\delta^{15}N$ of each sample and Eq. (1) to calculate $R_{sample}$
$(^{15}N/^{14}N)$. By defining $r_{sample}$ as $^{15}N/(^{14}N+^{15}N)$ in Eq. (2), we derived the $^{15}N$ and $^{14}N$
concentrations of all forms of N through Eqs. (3) and (4), with the exception of $NH_4^+$
and DON. The r value of the $NH_4^+$ was assumed to equal either its initial value or an
arbitrarily chosen fraction thereof, and the $^{15}N$ and $^{14}N$ content of the the $NH_4^+$ was then





determined using Eqs. (3) and (4). The $^{15}$N and $^{14}$N concent of the DON was then
determined by mass balance (N$_2$ fixation and emission of nitrogenous gases were
ignored).
$$\delta^{15}N\left(\text{\textperthousand}\,vs.air\right)=\left[\frac{\left(\frac{^{15}N}{^{14}N}\right)_{sample}}{\left(\frac{^{15}N}{^{14}N}\right)_{air}}-1\right]\times 1000 \tag{1}$$

$$r=\frac{^{15}N}{^{15}N+^{14}N}=\frac{\frac{^{15}N}{^{14}N}}{\frac{^{15}N}{^{14}N}+1} \tag{2}$$

$$\left[^{15}N\right]=\left[N\right]\times r \tag{3}$$

$$\left[^{14}N\right]=\left[N\right]\times(1-r) \tag{4}$$

**2.3 Formation of matrix equations**
Isotopic mass balance of the incubation system at every point in time was thus achieved.
In other words, the sums of the variations in the total N, $^{15}$N, and $^{14}$N concentrations
were zero, as shown in Eqs. (5–7).
$$\Delta\left[NH_4^+\right]+\Delta\left[NO_2^-\right]+\Delta\left[NO_3^-\right]+\Delta\left[PN\right]+\Delta\left[DON\right]=0 \tag{5}$$

$$\Delta\left[^{15}NH_4^+\right]+\Delta\left[^{15}NO_2^-\right]+\Delta\left[^{15}NO_3^-\right]+\Delta\left[^{15}\text{N-PN}\right]+\Delta\left[^{15}\text{N-DON}\right]=0 \tag{6}$$

$$\Delta\left[^{14}NH_4^+\right]+\Delta\left[^{14}NO_2^-\right]+\Delta\left[^{14}NO_3^-\right]+\Delta\left[^{14}\text{N-PN}\right]+\Delta\left[^{14}\text{N-DON}\right]=0 \tag{7}$$





In this newly proposed method, we added a tracer amount of $^{15}NH_4^+$ into the incubation
system at the very beginning and then monitored the changes of $^{15}N$ and $^{14}N$ in the five
pools. Subsamples were collected for analysis (at times of 0, 1.6, 4.4, 8.8, and 14.6 h in
high nutrient case and 0, 4.3, 11.8 and 23.7 in low nutrient case) after the start of the
experiment. We assumed no fractionation between $^{15}N$ and $^{14}N$ for all the transfer
reactions among the pools. The fluxes of $^{15}N$ and $^{14}N$ were therefore assumed to equal
the total flux multiplied by $r_{substrate}$ and $(1 - r_{substrate})$, respectively. Isotope fractionation
could easily be introduced into the equations if necessary, i.e. dividing $^{14}N$ flux by $\alpha$
(the ratio of specific rate constant of $^{14}N$ to $^{15}N$), and the flux of $^{15}N$ is obtained. In the
zero-order reaction scheme, the fluxes remain unchanged through time; however, the r
values for different pools may vary significantly due to the redistribution of the $^{15}N$
tracer. According to mass balance, the changes of the $^{15}N$ concentrations of the $NH_4^+$,
$NO_2^-$, $NO_3^-$, and (PN + DON) pools over time are determined by the inflow and
outflow of $^{15}N$, as shown by Eqs. (8–11), respectively. Because the DON release rate
($F_8$) is deduced from mass conservation, it is inappropriate to add the DON pool in the
matrix as an independent equation. Similarly, the temporal dependence of $^{14}N\text{-}NH_4^+$,
$^{14}N\text{-}NO_2^-$, $^{14}N\text{-}NO_3^-$ and $^{14}N\text{-}(PN+DON)$ were expressed by Eqs. (12–15),
respectively.
$$\frac{d\left[^{15}NH_4^+\right]}{dt} = F_2 \times r_{PN} - (F_1 + F_4) \times r_{NH_4^+} \tag{8}$$





$$\frac{d\left[ {}^{15}NO_2^- \right]}{dt} = F_4 \times r_{NH_4^+} + F_7 \times r_{PN} - (F_3 + F_6) \times r_{NO_2^-}$$ (9)
$$\frac{d\left[ {}^{15}NO_3^- \right]}{dt} = F_6 \times r_{NO_2^-} - F_5 \times r_{NO_3^-}$$ (10)
$$\frac{d\left[ {}^{15}N\text{-}PN \right]}{dt} + \frac{d\left[ {}^{15}N\text{-}DON \right]}{dt} = F_1 \times r_{NH_4^+} + F_3 \times r_{NO_2^-} + F_5 \times r_{NO_3^-} - F_2 \times r_{PN} - F_7 \times r_{PN}$$ (11)
$$\frac{d\left[ {}^{14}NH_4^+ \right]}{dt} = F_2 \times (1 - r_{PN}) - (F_1 + F_4) \times (1 - r_{NH_4^+})$$ (12)
$$\frac{d\left[ {}^{14}NO_2^- \right]}{dt} = F_4 \times (1 - r_{NH_4^+}) + F_7 \times (1 - r_{PN}) - (F_3 + F_6) \times (1 - r_{NO_2^-})$$ (13)
$$\frac{d\left[ {}^{14}NO_3^- \right]}{dt} = F_6 \times (1 - r_{NO_2^-}) - F_5 \times (1 - r_{NO_3^-})$$ (14)
$$\frac{d\left[ {}^{14}N\text{-}PN \right]}{dt} + \frac{d\left[ {}^{14}N\text{-}DON \right]}{dt} = F_1 \times (1 - r_{NH_4^+}) + F_3 \times (1 - r_{NO_2^-}) + F_5 \times (1 - r_{NO_3^-}) - F_2 \times (1 - r_{PN}) - F_7 \times (1 - r_{PN})$$

236 (15)

We solved Eqs. (8–15) for the fluxes $F_1$ through $F_7$ during the first two hours of the
experiment with the following matrix equation:






$$
\begin{pmatrix}
-r_{NH_4^+} & r_{PN} & 0 & -r_{NH_4^+} & 0 & 0 & 0 \\
0 & 0 & -r_{NO_2^-} & r_{NH_4^+} & 0 & -r_{NO_2^-} & r_{PN} \\
0 & 0 & 0 & 0 & -r_{NO_3^-} & r_{NO_2^-} & 0 \\
r_{NH_4^+} & -r_{PN} & r_{NO_2^-} & 0 & r_{NO_3^-} & 0 & -r_{PN} \\
-(1-r_{NH_4^+}) & (1-r_{PN}) & 0 & -(1-r_{NH_4^+}) & 0 & 0 & 0 \\
0 & 0 & -(1-r_{NO_2^-}) & (1-r_{NH_4^+}) & 0 & -(1-r_{NO_2^-}) & (1-r_{PN}) \\
0 & 0 & 0 & 0 & -(1-r_{NO_3^-}) & (1-r_{NO_2^-}) & 0 \\
(1-r_{NH_4^+}) & -(1-r_{PN}) & (1-r_{NO_2^-}) & 0 & (1-r_{NO_3^-}) & 0 & -(1-r_{PN})
\end{pmatrix}
\times
\begin{pmatrix}
F_1 \\ F_2 \\ F_3 \\ F_4 \\ F_5 \\ F_6 \\ F_7
\end{pmatrix}
=
\begin{pmatrix}
\frac{d[^{15}NH_4^+]}{dt} \\
\frac{d[^{15}NO_2^-]}{dt} \\
\frac{d[^{15}NO_3^-]}{dt} \\
\frac{d[^{15}N\text{-}PN]}{dt}+\frac{d[^{15}N\text{-}DON]}{dt} \\
\frac{d[^{14}NH_4^+]}{dt} \\
\frac{d[^{14}NO_2^-]}{dt} \\
\frac{d[^{14}NO_3^-]}{dt} \\
\frac{d[^{14}N\text{-}PN]}{dt}+\frac{d[^{14}N\text{-}DON]}{dt}
\end{pmatrix}
$$


241 (16)

The rates of change of the N concentrations were approximated by one-sided finite
difference expressions. For example, $d[^{14}NH_4^+]/dt$ at time t = 0 was appximated by
$\{[^{14}NH_4^+]_{t1} - [^{14}NH_4^+]_{t0}\}/2$ where the subscripts indicate the times at which the
concentrations were measured. Given these estimates of the derivatives on the
right-hand side of Eq. (16), we solved for the fluxes $F_1$ through $F_7$ during the first two
hours by assuming that the r values were equal to the average of the r values at times 0
and 2 hours. The flux $F_8$ was then determined by conservation of mass.
Unlike the high-nitrogen case, the reaction rate ($k_i*C$) changed over time as a result of
changes of the substrate concentration in the low-nitrogen scenario (Fig. 1b). The
relevant equations in that case are Eqs. (8–15). In this case nitrite and nitrate were
combined into one pool ($NO_x^-$). The total number of equations was therefore reduced
from eight to six. Meanwhile, $r_{substrate}$, $1 - r_{substrate}$ and $F_i$ were replaced by $[^{15}N]$, $[^{14}N]$




and $k_i$, respectively. Eq. (17) is the matrix form of the equations. Because in this case
the first-order reaction rates varied rapidly, short-term incubation data were especially
appropriate for calculating $k_i$ values. In solving Eq. (17) for the rate constants during
the first two hours of the experiment, we equated the $^{14}N$ and $^{15}N$ concentrations in the
left-hand matrix to the averages of the corresponding concentrations at $t0$ and $t1$.

$$\begin{pmatrix} -\left[^{15}NH_4^+\right] & \left[^{15}N\text{-}PN\right] & -\left[^{15}NH_4^+\right] & 0 & 0 \\ 0 & 0 & \left[^{15}NH_4^+\right] & -\left[^{15}NO_x^-\right] & \left[^{15}N\text{-}PN\right] \\ \left[^{15}NH_4^+\right] & -\left[^{15}N\text{-}PN\right] & 0 & \left[^{15}NO_x^-\right] & -\left[^{15}N\text{-}PN\right] \\ -\left[^{14}NH_4^+\right] & \left[^{14}N\text{-}PN\right] & -\left[^{14}NH_4^+\right] & 0 & 0 \\ 0 & 0 & \left[^{14}NH_4^+\right] & -\left[^{14}NO_x^-\right] & \left[^{14}N\text{-}PN\right] \\ \left[^{14}NH_4^+\right] & -\left[^{14}N\text{-}PN\right] & 0 & \left[^{14}NO_x^-\right] & -\left[^{14}N\text{-}PN\right] \end{pmatrix} \times \begin{pmatrix} k_1 \\ k_2 \\ k_3 \\ k_4 \\ k_5 \end{pmatrix} = \begin{pmatrix} \dfrac{d\left[^{15}NH_4^+\right]}{dt} \\ \dfrac{d\left[^{15}NO_x^-\right]}{dt} \\ \dfrac{d\left[^{15}N\text{-}PN\right]}{dt} + \dfrac{d\left[^{15}N\text{-}DON\right]}{dt} \\ \dfrac{d\left[^{14}NH_4^+\right]}{dt} \\ \dfrac{d\left[^{14}NO_x^-\right]}{dt} \\ \dfrac{d\left[^{14}N\text{-}PN\right]}{dt} + \dfrac{d\left[^{14}N\text{-}DON\right]}{dt} \end{pmatrix}$$

260                                                                                                  (17)

**2.4 Validation by Stella**
Because the matrix equations provide approximate solutions, we used STELLA 9.1.4
software (Isee systems, Inc.) to construct models that were consistent with the scenarios
depicted in Fig. 1 to simulate, as accurately as possible, the continuous fluxes of
nitrogen and to thus check the applicability of the isotope matrix method to analysis of
the observational data. The model was divided into two modules, one for $^{15}N$ and the
other for $^{14}N$. The modules balanced the total amounts of these isotopes in the $NH_4^+$,
$NO_2^-$, $NO_3^-$ (or $NO_x^-$), PN, and DON pools. The connection between these two
modules was through the $^{15}N$ atom % ($r_N$). The pool size was regulated by the F or k



values derived from solution of the matrix equations during the first two hours of the
experiments (Fig. S1 and S2). After setting the initial concentrations of $^{15}N$ and $^{14}N$ to
that measured in every pool, the model was run for 24 h according to the short-term F
(1.6 h) or k (4.3 h) values derived from the matrix equations. The model outputs of the
two cases are presented below. The output includes the time courses of the $^{15}N$ and $^{14}N$
concentrations and the $^{15}N$ atom% ($r_N$) or $\delta^{15}N$ of each N species. Through this
comparison, we could observe the evolution of the isotopic composition in the various
N pools.
In our case study, we measured all isotopic compositions, except that of $NH_4^+$. In the
cases presented below, we fixed the isotopic composition of $NH_4^+$ for the first model
run, i.e., no remineralization. This assumption has been made in many previous studies
(e.g. Dugdale and Goering, 1967; Ward et al., 1984; Santoro et al., 2010a, Santoro et al.,
2013; Hsiao et al., 2014; Peng et al., 2015). However, the assumption has been
criticised based on the fact that the labeled ammonium pool can be diluted by
regenerated ammonium (e.g. Caperon et al., 1979; Blackburn, 1979; Gilbert et al., 1982;
Dugdale and Wilkerson, 1986; Kanda et al., 1987; Dickson and Wheeler, 1995; Clark et
al., 2006; Raimbault and Garcia, 2008). The ammonium excretion rates calculated with
these dilution models, however, have been based on some assumptions. Examples of
these assumptions have included the following: (1) the uptake and excretion rates
remain constant during the incubation, (2) no $^{15}N$ is excreted, and (3) PN concentrations
change insignificantly. In addition, the amounts of $^{15}NH_4^+$ that needed to be added have



sometimes been greater than the ambient concentration, depending on the number of
trophic levels in the system (Caperon et al., 1979; Blackburn, 1979; Kanda et al., 1987).
All of the models have assumed that PON was the only sink of $^{15}N$. These studies have
broadened our insight into $NH_4^+$ cycling, even though they have made arbitrary
assumptions and considered only $NH_4^+$ excretion and incorporation into PN. To test the
validity of the assumption of insignificant $NH_4^+$ excretion in our cases, we activated
remineralization to various degrees in the model runs. We also compared the observed
and remineralization-associated simulations.
**3. High-nutrient case in a coastal bay in southern China**
**3.1 Study site and environmental data**
Wuyuanwan (WYW), located at the southern coast of China, is an inner bay with a
regular semidiurnal tide. Its water flows out during neap tide, and water from the open
ocean flows into the bay during spring tide. The water is well ventilated and constantly
saturated with dissolved oxygen. As a coastal bay, Wuyuanwan suffers from
anthropogenic influences that result in high nutrient concentrations analogous to other
coastal zones in China. It is an ideal research site to study the dynamic transformation
processes of the coastal nitrogen cycle.
During our sampling (January 2014), the water was vertically well mixed, with a
temperature of ~13.7 °C, a salinity of approximately 29.5, and pH of 8.1–8.3. The
concentrations of nitrogenous species were relatively high in this highly eutrophic





aquatic system, with inorganic nutrient concentrations of $30.9 \pm 0.7$ $\mu$mol L$^{-1}$ for NO$_3^-$,
$22.3 \pm 4.3$ $\mu$mol L$^{-1}$ for NH$_4^+$, $5.5 \pm 0.1$ $\mu$mol L$^{-1}$ for NO$_2^-$, and $8.5 \pm 0.2$ $\mu$mol L$^{-1}$ for
PN.
**3.2 Incubation experiments**
Water samples were collected in two pre-washed 10-L polycarbonate bottles (Nalgene,
USA). The sampling depth was 0.3 m, with a light intensity of 80 % of the surface water
irradiance. $^{15}$N-labeled NH$_4$Cl (98 atom % $^{15}$N, Sigma-Aldrich, USA) was added to the
incubation bottles (< 10 % of the ambient concentration). The incubations were carried
out immediately in an incubator equipped with a light screen allowing 80% light
penetration. The temperature was maintained at ~13.7 °C using continuously pumped
seawater. The first sample ($t_0$) was taken immediately after adding the tracer.
Subsequent samples were taken at approximately 2–4 h intervals. An aliquot of 200 mL
was filtered through a 47-mm polycarbonate membrane with a 0.22 $\mu$m pore size
(Millipore, USA), and the filtrates were frozen at –20 °C for chemical analysis in the
lab. Particulate matter was collected by filtering seawater through pre-combusted
(450 °C for 4 h) GF/F filters that were 25 mm in diameter (Whatman, GE Healthcare,
USA), under a pressure of <100 mm Hg. The GF/F filters were freeze-dried and stored
in a desiccator for further analyses of the PN concentration and isotopes. We selected
the first 16 hours for presentation here.
**3.3 Results**



### 331  3.3.1 Observational results

The concentrations, nitrogen isotope signatures, $^{15}$N atom percentages, and $^{15}$N and $^{14}$N
concentrations of $NH_4^+$, $NO_2^-$, $NO_3^-$, PN, and DON in the incubation showed
distinctive patterns along with the incubation time (Fig. 2). The concentrations of $NH_4^+$
and $NO_3^-$ were higher than those of $NO_2^-$, PN, and DON. $NH_4^+$ significantly and
continuously decreased from 26.6 to 16.5 $\mu$mol L$^{-1}$ at a rate of 0.63 $\mu$mol L$^{-1}$ h$^{-1}$ over
the course of the incubation (Fig. 2a). $NO_3^-$ decreased from 30.9 to 28.3 $\mu$mol L$^{-1}$ at a
rate of approximately one-third the rate of $NH_4^+$ (Fig. 2c). The $NO_2^-$ concentration
displayed a slightly declining trend (Fig. 2b). Conversely, the PN and DON
concentrations steadily increased. The PN concentration increased from 8.8 to 18.3
$\mu$mol L$^{-1}$ at a rate of 0.66 $\mu$mol L$^{-1}$ h$^{-1}$, which was very close to that of $NH_4^+$ (Fig. 2d).
The DON concentration increased from 17.4 to 20.9 $\mu$mol L$^{-1}$ at a rate of 0.22 $\mu$mol L$^{-1}$
h$^{-1}$ (Fig. 2e).
The time courses of the nitrogen isotopic compositions of $NH_4^+$, $NO_2^-$, $NO_3^-$, PN and
DON in the incubation are shown in Figs 2 f–j. The $\delta^{15}$N-$NH_4^+$ value remained constant
without considering $NH_4^+$ regeneration (Fig. 2f), and $\delta^{15}$N-$NO_2^-$ increased from –9.0 to
12.1 ‰ (Fig. 2g); $\delta^{15}$N-$NO_3^-$ ranged from 6.9 to 9.9 ‰ with no significant trend over
time (Fig. 2h). In addition, $\delta^{15}$N-PN increased from 14.8 to 2718.8 ‰ (Fig. 2i). Based
on the N mass and isotope balance, the calculated value of $\delta^{15}$N-DON followed the
same trend as $\delta^{15}$N-PN, increasing from 5.0 to 2617.6 ‰ (Fig. 2j). According to the
$\delta^{15}$N values of these N pools, the $^{15}$N atom percentages ($r_N$) were calculated; they





displayed a similar pattern as the corresponding $\delta^{15}N$ variation (Figs. 2 k–o). The r
values of $NO_2^-$ and $NO_3^-$ varied within narrow ranges (0.36–0.37 %), ten times less
than those of $NH_4^+$, PN, and DON.
The $^{15}N$ and $^{14}N$ concentrations of $NH_4^+$, $NO_2^-$, $NO_3^-$, PN, and DON were computed
(Figs. 2 p–y) based on their bulk concentrations and $^{15}N$ atom percentages ($r_N$). The
$^{15}N$-$NH_4^+$ and $^{14}N$-$NH_4^+$ concentrations decreased significantly (Figs. 2 p and u) at rates
of 0.026 and 0.61 $\mu$mol $L^{-1}$ $h^{-1}$, respectively. The $^{15}N$-$NO_2^-$ and $^{15}N$-$NO_3^-$
concentrations varied within relatively small ranges from 0.020 to 0.019 $\mu$mol $L^{-1}$ and
from 0.11 to 0.10 $\mu$mol $L^{-1}$, respectively (Figs. 2 q and r). The $^{14}N$-$NO_2^-$ and $^{14}N$-$NO_3^-$
concentrations declined significantly (Figs. 2 v and w) compared with the $^{15}N$-$NO_2^-$
and $^{15}N$-$NO_3^-$ concentrations. In contrast, the $^{15}N$-PN and $^{14}N$-PN concentrations
increased remarkably (Figs. 2 s and x); the $^{15}N$-DON and $^{14}N$-DON concentrations
exhibited increasing trends similar to that of PN (Figs. 2 t and y). In fact, many of the
previous incubation studies observed a significant nitrogen imbalance that was
attributed to DON release, reaching an average of ~10–45% (e.g. Dugdale and
Wilkerson, 1986; Ward and Bronk, 2001; Bronk and Ward, 1999, 2005; Varela et al.,
2005). This magnitude of DON release is in line with our incubation results.
**3.3.2 Solutions of the matrix equation and STELLA back calculation**
As presented above, the bulk, $^{15}N$ and $^{14}N$ concentrations of $NH_4^+$, $NO_2^-$, $NO_3^-$, PN, and
DON varied linearly with incubation time, indicating that the reactions in the



incubation system could be treated as a zero-order reaction. We selected the first
sampling interval for the calculation of rate constants, although the incubation was
conducted for 14.6 h. The r values ($NH_4^+$, $NO_2^-$, $NO_3^-$, and PN) used for computation
were the average values of the corresponding pool in the first incubation interval. The
results under the assumption of fixed $r_{NH4+}$ conditions are shown in Table 1. The $NH_4^+$
uptake rate ($F_1$), 0.63 $\mu$mol $L^{-1}$ $h^{-1}$, was much higher than the other rates and followed
by the $NO_3^-$ uptake rate ($F_5$, 0.22 $\mu$mol $L^{-1}$ $h^{-1}$) and DON release rate ($F_8$, 0.25 $\mu$mol $L^{-1}$
$h^{-1}$). The $NO_2^-$ uptake ($F_3$) rate was 0.032 $\mu$mol $L^{-1}$ $h^{-1}$, much lower than that of $NH_4^+$
and $NO_3^-$ uptake. The ammonia oxidation rate ($F_4$) was 0.00090 $\mu$mol $L^{-1}$ $h^{-1}$, but the
nitrite oxidation rate ($F_6$) was undetectable. Since the incubation was conducted under
80 % light conditions, low rates of ammonium and nitrite oxidation were reasonable
because either nitrifiers and NOB are sensitive to light (e.g. Olson, 1981a, 1981b;
Horrigan et al., 1981; Ward, 2005; Merbt et al., 2012). In addition, the nitrification rates
may have been constrained by competition with phytoplankton for ammonium under
the relatively strong light field (Smith et al., 2014). The nitrite release rate by
phytoplankton ($F_7$) was nearly zero. Similarly, nitrite release was below the limit of
detection in Monterey Bay observed by Santoro et al (Santoro et al., 2013).
By introducing the measured initial [15]N and [14]N concentrations of $NH_4^+$, $NO_2^-$, $NO_3^-$,
PN, and DON and the calculated rates ($F_1$–$F_8$) into the STELLA model (Fig. S1), we
obtained successive variations of [15]N and [14]N concentrations and $r_N$ of $NH_4^+$, $NO_2^-$,
$NO_3^-$, PN, and DON over time (Figs. 3 a–o). The model output of the $\delta^{15}$N values and



bulk N concentrations of these N species could thus be derived (Figs. 3 p–y). The
modeled and measured values remained consistent throughout the incubation.
To test the effect of regeneration (i.e., activating regeneration and $F_2 > 0$), we allowed
$r_{NH4+}$ to decrease to different degrees (1%, 10%, 20%, and 50% of the total at the end of
the incubation). As indicated in previous studies, such regeneration-induced isotope
dilution indeed altered the original results (Fig. 3). Since ammonium uptake is the
dominant process, the alteration of the PN pool was more significant in comparison
with the other pools (Figs. 3 d, n and s). To maintain a constant reduction of the
measured $NH_4^+$ concentration, $F_1$ increased as $F_2$ increased (Table 1). As the
regeneration increased, the deviation of the time course of $^{15}N$-PN production (Fig. 3c)
increased, resulting in a larger curvature of r-PN and $\delta^{15}N$-PN, and the turning point
appeared earlier. This model exercise confirmed the influence of the isotope dilution
effect; however, this effect is insignificant in the very early stage of an incubation. Such
a result suggests that a better result can be obtained in a short-term incubation only
when regeneration is intensive. Nevertheless, the matrix solution fit well with the
model run with fixed r-$NH_4^+$, suggesting that the assumption of no regeneration was
plausible, at least in our case during the incubation period.
**4. Low-nutrient case in the western North Pacific (WNP)**
**4.1 Sampling station and incubation experiment**





412 The WNP cruise took place from 30 March to 5 May in 2015 aboard the R/V

413 Dongfanghong 2. The survey area covered 25 to 32° N and 120 to 152° E.

414 The station for the experiment was located at 32°37.838′ N and 145°56.759′ E, and

415 water samples were collected using a 24-bottle rosette sampler. The sampling depth

416 was 25 m with relatively low light intensity. Pre-washed 10-L polycarbonate carboys

417 (Nalgene, USA) were used for the incubation. A total 1.5 mL of 200 $\mu$M $^{15}$N-labelled

418 NH$_4$Cl tracers containing 98 atom% $^{15}$N (Sigma-Aldrich, USA) was injected into the

419 incubation bottle to achieve a final concentration of 30 nM. Incubation was carried out

420 immediately in a thermostatic incubator (GXZ-250A, Ningbo) with a constant light

421 intensity in 33 % level (300 Lux on average) at 18.4 °C, the same as the *in situ* sampling

422 temperature.

423 **4.2 Results**

424 **4.2.1 Observational results**

425 The patterns of the variations of the bulk N concentration, nitrogen isotope signature,

426 $^{15}$N atom percentage, and $^{15}$N and $^{14}$N concentrations of the NH$_4^+$, NO$_x^-$, PN, and DON

427 during the incubation are shown in Fig. 4. DON was the dominant N pool (several tens

428 of times higher than NH$_4^+$, NO$_x^-$, and PN). NH$_4^+$ and NO$_x^-$ decreased rapidly at the

429 early stage and later more slowly, dropping from 142.7 to 48.1 nM and 520.7 to 126.8

430 nM, respectively (Figs. 4 a and b). In contrast, PN and DON continuously increased.





The PN concentration increased from 436.8 to 667.0 nM (Fig. 4c), and the DON
concentration increased from 5.4 to 5.6 $\mu$M (Fig. 4d).
Opposite to the trend of $NO_x^-$ concentrations, $\delta^{15}N\text{-}NO_x^-$ increased from 8.9 to 170.6 ‰
(Fig. 4f). In addition, $\delta^{15}N\text{-}PN$ exhibited great changes, increasing from 47 to 6948 ‰
(Fig. 4g). Based on the N mass and isotope balance, the calculated $\delta^{15}N\text{-}DON$ generally
showed an increasing trend from 5.0 to 150.1 ‰ that was quite rapid in the beginning
and later became steady (Fig. 4h). The variation of $^{15}N$ atom percentage ($r_N$) presented a
very similar, but not as significant, trend as the corresponding $\delta^{15}N$ variation of specific
N species (Figs. 4 i–l). The r values of $NO_x^-$ and DON varied over a narrow range from
0.36 to 0.42% (Figs 4 j and l), and $r_{PN}$ increased from 0.38 to 2.83 % (Fig. 4k).
The $^{15}N$ and $^{14}N$ concentrations of $NH_4^+$, $NO_x^-$, PN, and DON (Figs. 4 m–t) were
computed, and their trends over time resembled the corresponding bulk concentration
changes. The $^{15}N\text{-}NH_4^+$ and $^{14}N\text{-}NH_4^+$ concentrations decreased from 29.8 to 10.0 nM
and from 112.9 to 38.0 nM, respectively (Figs. 4 m and q). The $^{15}N\text{-}NO_x^-$ declined from
1.9 to 0.5 nM, and $^{14}N\text{-}NO_x^-$ decreased from 518.8 to 126.3 nM (Figs. 4 n and r). In
contrast, the $^{15}N\text{-}PN$ and $^{14}N\text{-}PN$ concentrations increased notably from 1.7 to 18.9 nM
and 435.1 to 648.1 nM, respectively (Figs. 4 o and s). The $^{15}N$ and $^{14}N$ concentrations of
DON showed an increasing trend similar to that of PN, ranging from 19.8 to 23.7 nM
and from 5.4 to 5.6 $\mu$M, respectively (Figs 4 p and t).
**4.2.2 Solutions of the matrix equation and STELLA back calculation**



The temporal variations of the bulk, $^{15}$N, and $^{14}$N concentrations of $NH_4^+$, $NO_x^-$, PN,
and DON with incubation time revealed patterns of exponential decrease or increase
(Fig. 4), demonstrating that the reactions in this low-nutrient system could be treated as
first-order reactions. As first-order reactions, the specific rates were fixed. Here, we
chose the first sampling interval, i.e., 4.3 h, for the specific rate calculation. The $^{15}$N and
$^{14}$N concentrations of $NH_4^+$, $NO_x^-$, and PN for matrix computation were the mean
values for the specific time interval.
The results under the assumption of constant $r_{NH4+}$ are shown in Table 2. The $NO_x^-$
specific uptake rate ($k_4$) was 0.059 h$^{-1}$ (27.12 nmol L$^{-1}$ h$^{-1}$), the highest among these
reaction rates, followed by the $NH_4^+$ specific uptake rate ($k_1$, 0.045 h$^{-1}$) and nitrification
specific rate ($k_3$, 0.00050 h$^{-1}$). The specific rate of the release of $NO_2^-$ by phytoplankton
($k_5$) was undetectable, similar to a previous report by Santoro et al. (2013), who
suggested that phytoplankton $NO_2^-$ excretion may only occur under Fe-limited
conditions or when phytoplankton rely on a single N source.
By introducing the measured initial $^{15}$N and $^{14}$N concentrations of $NH_4^+$, $NO_x^-$, PN, and
DON and the calculated specific rates ($k_1$ to $k_5$) into STELLA (Fig. S2), we obtained
consecutive changes in all parameters (see Fig. 5). Generally, the model outputs fit well
with the measured values, except for the $^{15}$N concentration, $\delta^{15}$N, and $r_N$ of PN, for
which the last data points were slightly higher than the model (Figs. 5 c, k and o). Since
this case was conducted under low-nutrient conditions, this positive offset was





probably compensated for by organic nitrogen utilization. The low level of nitrate and
ammonium, in fact, was approaching the concentration threshold for phytoplankton
utilization (e.g., <30–40 nM $NH_4^+$ for *Emiliania huxleyi*; Sunda and Ransom, 2007).
Our flow cytometry data demonstrated that the growth of eukaryotes was slowing down,
but the growth of *Synechococcus* was continuous (not shown). In general, the system
followed a first-order reaction in the first 16 hours, and this simulation demonstrated
the applicability of this matrix method.
In order to test the validity of the assumption of no regeneration, $r_{NH4+}$ was artificially
decreased by 1 %, 10 %, 20 %, and 50 % in total by the end of the incubation (a span of
23.7 h). The experiment was conducted in the same way as the high-nutrient case, and
the results are shown in Table 2 and Fig. 5. The specific $NH_4^+$ uptake rate ($k_1$) increased
as the regeneration ($k_2$) increased (Table 2). This resut demonstrated that the effect of
$r_{NH4+}$ on $NO_x^-$-associated parameters was trivial (Figs. 5 b, f, j, n, and r). In contrast, the
variation of $NH_4^+$ regeneration significantly affected the $^{15}N$ concentration, $\delta^{15}N$, and
$r_N$ of PN. More specifically, greater $NH_4^+$ regeneration resulted in larger differences
between these three PN-associated values and the STELLA-modeled data (Figs. 5 c, k
and o). Thus, the $NH_4^+$ regeneration rate could be ignored, at least in the early stage in
our case, a reflection of the good agreement with the model run.
**5. Comparisons with traditional methods**





Below, we present a comparison with conventional rate measurements of ammonium
oxidation and uptake. The most popular N uptake rate calculation (i.e. Eq. (18)) follows
Dugdale and Wilkerson (1986), and similarly, the calculation of ammonium oxidation
in Eq. (19):
$$\rho_{NH_4^+} = \frac{(r_t - r_0) \times [PN]_t}{(r_{NH_4^+} - r_0) \times T},$$    (18)
$$\rho_{NR} = \frac{(r_t - r_0) \times [NO_X^-]_t}{(r_{NH_4^+} - r_0) \times T},$$    (19)
where $\rho_{NH4+}$ ($\rho_{NR}$) stands for the $NH_4^+$ assimilation (oxidation) rate; $r_0$ and $r_t$
represent the initial and final $^{15}N$ atom %, respectively, in the PN ($NO_2^-/NO_x^-$) pool;
$r_{NH4+}$ is the initial $^{15}N$ atom % of $NH_4^+$; $[PN]_t$ ($[NO_x^-]$) represents the final PN ($[NO_x^-]$)
concentration; and T is the incubation time in hours.
The end products in the above equations, in fact, were also influenced by non-target
processes. For example, the DON release (regardless of its cause) was not considered
in the canonical method for ammonium uptake (Eq. (18) considers only the $^{15}N$
retained in particulate pool). In Table 3, the $NH_4^+$ uptake rate calculated by the matrix
method (632.2 nmol $L^{-1}$ $h^{-1}$) was ~53 % higher than the rate calculated by the
traditional method (413.6 nmol $L^{-1}$ $h^{-1}$) for the high-nutrient case in WYW. In the
low-nutrient case, the matrix-derived $NH_4^+$ uptake rate (4.58 nmol $L^{-1}$ $h^{-1}$) was ~20 %
higher than that (3.86 nmol $L^{-1}$ $h^{-1}$) from the traditional method. The higher uptake rates
were mainly due to the DON release, which was not counted in the traditional method.





Whether the uncounted DON portion was caused by active or passive release (e.g.,
sloppy feeding, cell death, viral infection, or physiological limitation) remains unclear,
and the magnitude of DON release relative to the total N uptake remains undetermined,
even though a handful of studies have revealed its significance (Bronk and Glibert,
1993; Bronk and Steinberg, 2008; Sipler and Bronk, 2015). More experiments are
required to explore the importance of this uncounted portion in field studies.
Nevertheless, our results from the mass balance and matrix solution suggest the
significance of DON release, regardless of the nutrient status.
On the other hand, the end products of ammonium oxidation or nitrification are
consumed by phytoplankton continuously, particularly in a euphotic layer full of
photosynthetic autotrophs. In many cases, nitrate uptake occurs in both light and dark
conditions (e.g. Dugdale and Goering, 1967; Lipschultz, 200); Mulholland and Lomas,
2008). The significant consumption of end products ($NO_2^-$ and $NO_x^-$) may bias the
conventional rate calculation. We can clearly see that possibility in Eq. (19). Even
though the consumption of $NO_x^-$ results in a net decreasing trend of $^{15}NO_x^-$ (Figs. 2r
and 4n), the $NH_4^+$ oxidation/nitrification rate in both the WYW and WNP cases could
be obtained (see Table 3) as long as $r_{NOx-}$ was increasing (i.e., positive ($r_t - r_0$) in Eq.
(19)). The $NH_4^+$ oxidation/nitrification rate in the high- and low-nutrient cases (0.41
and 0.046 nmol $L^{-1}$ $h^{-1}$, respectively) derived from the canonical method were lower
than those (0.90 and 0.051 nmol $L^{-1}$ $h^{-1}$, respectively) from the matrix method. This
apparent discrepancy resulted from product consumption. In fact, Santoro et al. (2010a,



2013) realized the effect of consumption on their rate calculation. To overcome this
consumption effect induced by the first-order reaction, they took $NO_x^-$ removal into
consideration and formulated a new equation, a function of nitrification rate (F) and
$NO_x^-$ uptake rate (k). Following Santoro et al. (2010a), we calculated the nitrification
rate for the low-nutrient case (via a nonlinear least-squares curve-fitting routine in
Matlab by using the first three time points of the $^{15}N_{NOx-}/^{14}N_{NOx-}$ measurements) to be
0.056 nmol $L^{-1} h^{-1}$ (Table 3), which was slightly (~10%) larger than the matrix-derived
rate (0.051 nmol $L^{-1} h^{-1}$). The simulations of $\delta^{15}NO_x^-$ and $r_{NOx-}$ deduced from results by
the method of Santoro et al. (2010a) agreed well with the isotope matrix method (Figs.
5 j and n). Interestingly, their nitrate uptake rate (k = 0.010 $h^{-1}$) was only one-sixth that
(0.059 $h^{-1}$) derived from the matrix method, although a comparable nitrification rate
was obtained when the consumption term was taken into account. Surprisingly, when
we introduced the two parameters to generate the time courses of the $^{15}NO_x^-$, $^{14}NO_x^-$,
and $NO_x^-$ concentrations, we found much slower decreasing trends in the
concentrations (Figs. 5 b, f, and r). In fact, the formula produced by Santoro et al.
(2010a) is constrained only by the ratio changes rather than the individual
concentration changes in $^{15}NO_x^-$ and $^{14}NO_x^-$. Thus, the nonlinear curve-fitting method
by Matlab may only provide a correct simulation for the ratio change. This implies that
the nitrate uptake rate derived from the non-linear curve-fitting method in Matlab
should be validated by using the concentration of nitrate at the end point, as was done





by Santoro et al (2013). Thus, a precise measurement of concentration changes is vital
in time series incubations for nitrification.
To evaluate the fractional contribution of nitrification to   $NO_3^-$ uptake as done by Yool
et al. (2007), labeled $^{15}NH_4^+$ and $^{15}NO_3^-$ addition were needed in parallel incubations;
meanwhile, a realistic evaluation can be achieved only when the incubation is
conducted in the same bottle under *in situ* light conditions, in which light inhibition
and substrate competition must occur simultaneously. The isotope matrix method is so
far the most convenient and suitable method for evaluating the relative importance of
co-occuring nitrification and new production. Through the matrix, the contributions of
nitrification to new production were approximately 0.4 % and 0.2 % in the high- and
low-nutrient cases, respectively; these relatively low values were probably due to the
light inhibition effects on nitrifiers.
Another example is resolution of the mechanisms of formation of the primary nitrite
maximum (PNM). Previous studies have involved addition of various tracers into
parallel incubation bottles to determine associated individual processes (Olson, 1981a,
1981b; Lomas and Lipschultz, 2006; Santoro et al., 2013). However, this laboursome
operation does not exclude the dynamic N interactions. Furthermore, the matrix
method is also appropriate for probing the effects of environmental factors (e.g., $CO_2$,
pH, temperature, light intensity, and dissolved oxygen) on the interactive N processes
in one single incubation bottle. For example, after synthesizing studies of $CO_2$ effects



on $N_2$-fixation, nitrification, and denitrification, Hutchins et al. (2009) indicated that
the N cycle may strongly respond to higher $CO_2$. By labeling one N species and
controlling the level of $CO_2$, our isotope matrix method can determine these rates
simultaneously. Thus, a better evaluation of the response of the N cycle to rising $CO_2$
can be achieved.
**6. Conclusion**
Although the assessment of relevant errors is weakened due to the involvement of
different error sources (analytical error, error propagation in calculation, and matrix
solution error), and the estimate of uncertainty for this isotope matrix method is not a
simple statistical question, the isotope matrix method saves both labor and time in the
field if one intends to obtain multiple rates simultaneously. Given the progress in
analytical techniques used to measure the concentration and isotopic composition of
nitrogen species, the isotope matrix method presents a promising avenue for the study
of rates of nitrogen processes with a system-wide perspective.
**Acknowledgement**
We sincerely thank Wenbin Zou and Tao Huang at the State Key Laboratory of Marine
Environmental Science (Xiamen University, China) for their valuable help with the
water sampling and the on-board trace $NH_4^+$ concentration analysis during the 2015
NWP cruise. This research was funded by the National Natural Science Foundation of
China (NSFC U1305233, 2014CB953702, 91328207, 2015CB954003).





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





**Figure Captions**

**Fig. 1.** Model schemes with well-recognized nitrogen transformation processes in high-
(a) and low- (b) nutrient aquatic environments. Two pools (inorganic and organic) were
categorized (see text). Arrows stand for the transfer flux/rate from the reactant to
product pool. The structure and inter-exchanges in the low-nutrient case (Fig. 1 b) are
the same as in (a), except that $NO_2^-$ and $NO_3^-$ are combined into $NO_x^-$.

**Fig. 2.** Time courses of (a) $[NH_4^+]$, (b) $[NO_2^-]$, (c) $[NO_3^-]$, (d) [PN], (e) [DON], (f)
$\delta^{15}N\text{-}NH_4^+$, (g) $\delta^{15}N\text{-}NO_2^-$, (h) $\delta^{15}N\text{-}NO_3^-$, (i) $\delta^{15}N\text{-}PN$, (j) $\delta^{15}N\text{-}DON$, (k) $r_{NH4+}$, (l)
$r_{NO2-}$, (m) $r_{NO3-}$, (n) $r_{PN}$, (o) $r_{DON}$, (p) $[^{15}NH_4^+]$, (q) $[^{15}NO_2^-]$, (r) $[^{15}NO_3^-]$, (s) $[^{15}N\text{-}PN]$, (t)
$[^{15}N\text{-}DON]$, (u) $[^{14}NH_4^+]$, (v) $[^{14}NO_2^-]$, (w) $[^{14}NO_3^-]$, (x) $[^{14}N\text{-}PN]$ and (y) $[^{14}N\text{-}DON]$
during the incubation in the high nutrient case. Data in the plots with grey backgrounds
were obtained by mass conservation under the assumption of no $NH_4^+$ regeneration.

**Fig. 3.** STELLA-derived and observed values in the high-nutrient case for (a) $[^{15}NH_4^+]$,
(b) $[^{15}NO_2^-]$, (c) $[^{15}NO_3^-]$, (d) $[^{15}N\text{-}PN]$, (e) $[^{15}N\text{-}DON]$, (f) $[^{14}NH_4^+]$, (g) $[^{14}NO_2^-]$, (h)
$[^{14}NO_3^-]$, (i) $[^{14}N\text{-}PN]$, (j) $[^{14}N\text{-}DON]$, (k) $r_{NH4+}$, (l) $r_{NO2-}$, (m) $r_{NO3-}$, (n) $r_{PN}$, (o) $r_{DON}$, (p)
$\delta^{15}N\text{-}NH_4^+$, (q) $\delta^{15}N\text{-}NO_2^-$, (r) $\delta^{15}N\text{-}NO_3^-$, (s) $\delta^{15}N\text{-}PN$, (t) $\delta^{15}N\text{-}DON$, (u) $[NH_4^+]$, (v)
$[NO_2^-]$, (w) $[NO_3^-]$ (x) [PN] and (y) [DON]. The black open triangles represent the
observational values; the black solid line indicate the STELLA model simulation
under constant $r_{NH4+}$; and the green, blue, magenta and pink lines represent the
simulations of 1%, 10%, 20% and 50% decreases in $r_{NH4+}$, respectively.





**Fig.4.** Time courses of (a) $[NH_4^+]$, (b) $[NO_x^-]$, (c) [PN] (d) [DON], (e) $\delta^{15}N\text{-}NH_4^+$, (f)
$\delta^{15}N\text{-}NO_x^-$, (g) $\delta^{15}N\text{-}PN$, (h) $\delta^{15}N\text{-}DON$, (i) $r_{NH4+}$, (j) $r_{NOx-}$, (k) $r_{PN}$, (l) $r_{DON}$, (m)
$[^{15}NH_4^+]$, (n) $[^{15}NO_x^-]$, (o) $[^{15}N\text{-}PN]$, (p) $[^{15}N\text{-}DON]$, (q) $[^{14}NH_4^+]$, (r) $[^{14}NO_x^-]$, (s)
$[^{14}N\text{-}PN]$ and (t) $[^{14}N\text{-}DON]$ during the incubation in the low-nutrient case. Data in the
plots with grey backgrounds were obtained under the assumption of no $NH_4^+$
regeneration. Error bars were shown also in plots and in many cases errors are smaller
than the size of the symbols.
**Fig. 5.** STELLA-derived and observed values in the low-nutrient case for (a) $[^{15}NH_4^+]$,
(b) $[^{15}NO_x^-]$, (c) $[^{15}N\text{-}PN]$, (d) $[^{15}N\text{-}DON]$, (e) $[^{14}NH_4^+]$, (f) $[^{14}NO_x^-]$, (g) $[^{14}N\text{-}PN]$, (h)
$[^{14}N\text{-}DON]$, (i) $r_{NH4+}$, (j) $r_{NOx-}$, (k) $r_{PN}$, (l) $r_{DON}$, (m) $\delta^{15}N\text{-}NH_4^+$, (n) $\delta^{15}N\text{-}NO_x^-$, (o)
$\delta^{15}N\text{-}PN$, (p) $\delta^{15}N\text{-}DON$, (q) $[NH_4^+]$, (r) $[NO_x^-]$, (s) [PN] and (t) [DON]. The black
open triangles represent the observed values; the black solid lines indicate the
STELLA model simulation under constant $r_{NH4+}$; and the green, blue, magenta and
pink lines stand for simulations of 1%, 10%, 20% and 50% decreases in $r_{NH4+}$,
respectively. The dashed lines in (b), (f), (j), (n) and (r) were generated from nonlinear
least-squares curve-fitting by Matlab following Santoro et al. (2010).





**Table Captions**
**Table 1.** The matrix results for the rates of N processes in the high-nutrient case under
different $r_{NH4+}$ variation conditions.
**Table 2.** The matrix results for the specific rates of N processes in the low-nutrient case
during the time series incubation.
**Table 3.** Comparison of the $NH_4^+$/ $NO_x^-$ uptake and $NH_4^+$ oxidation/nitrification rate
calculations by the matrix and conventional methods.



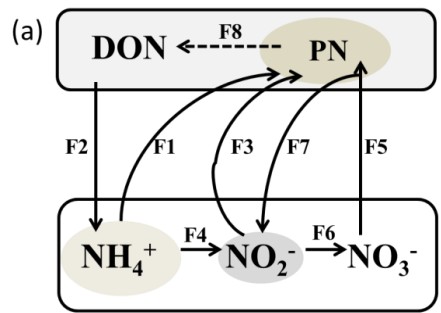 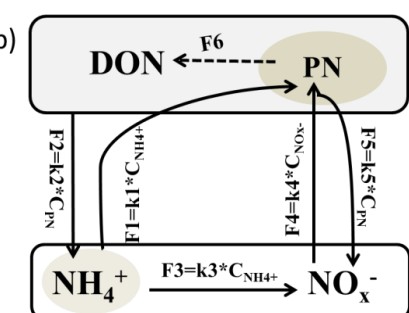



**Figure 1**



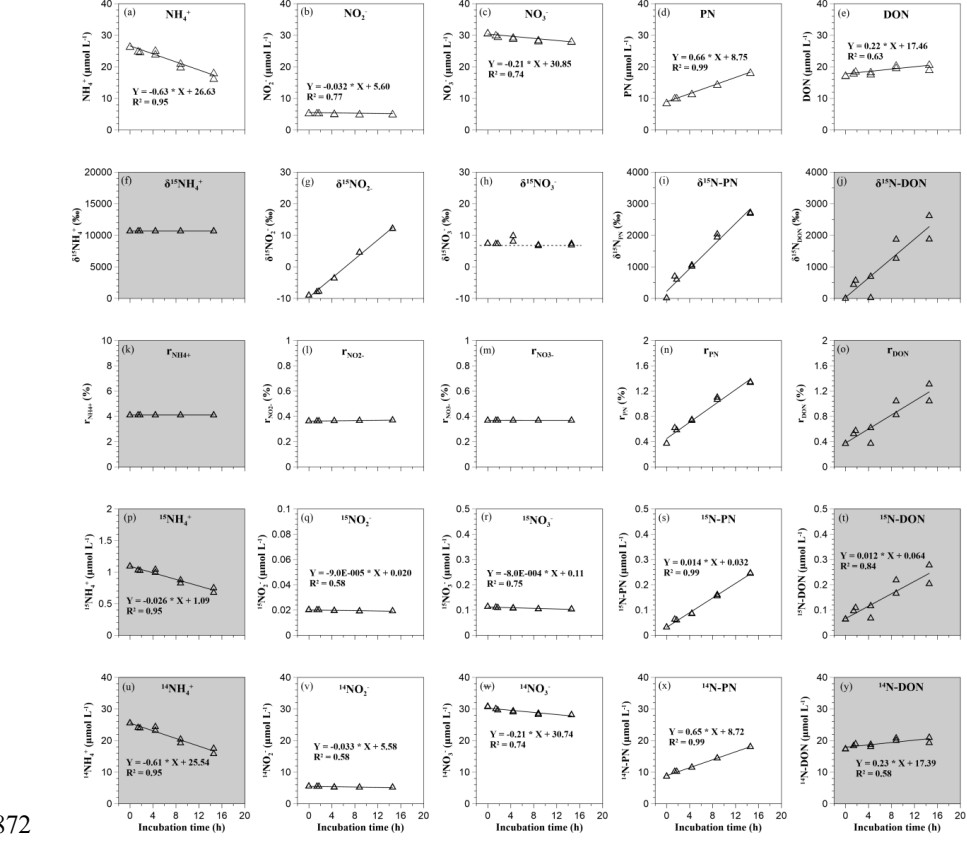



**Figure 2**



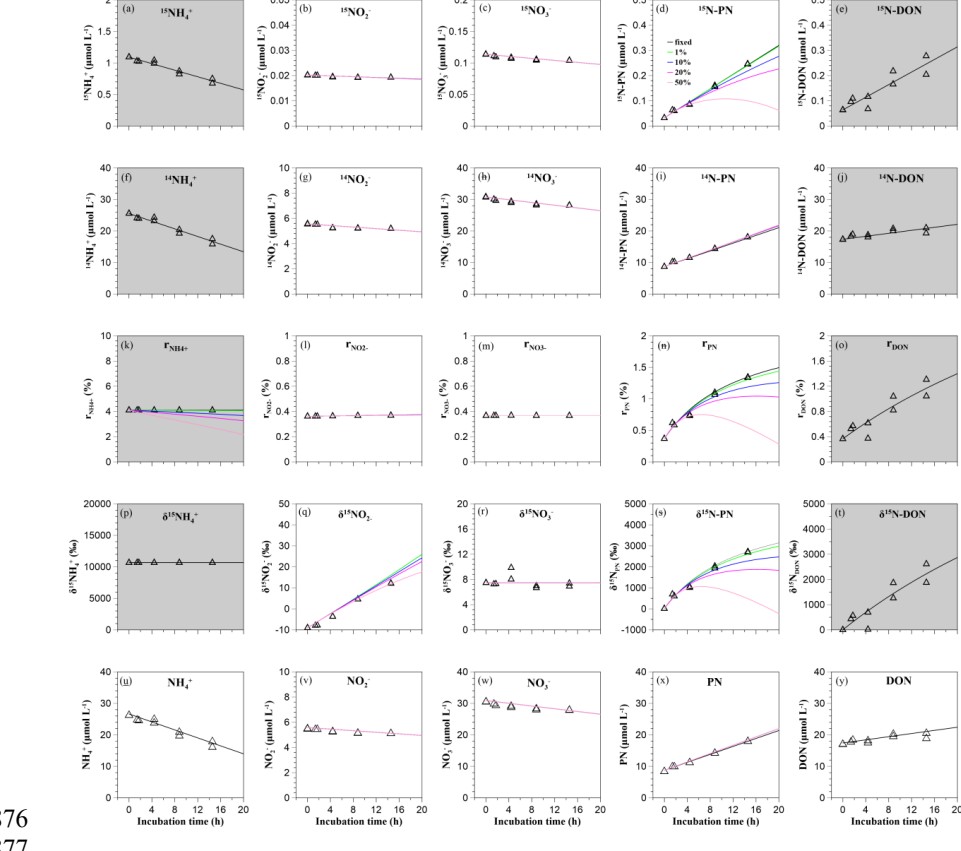



**Figure 3**






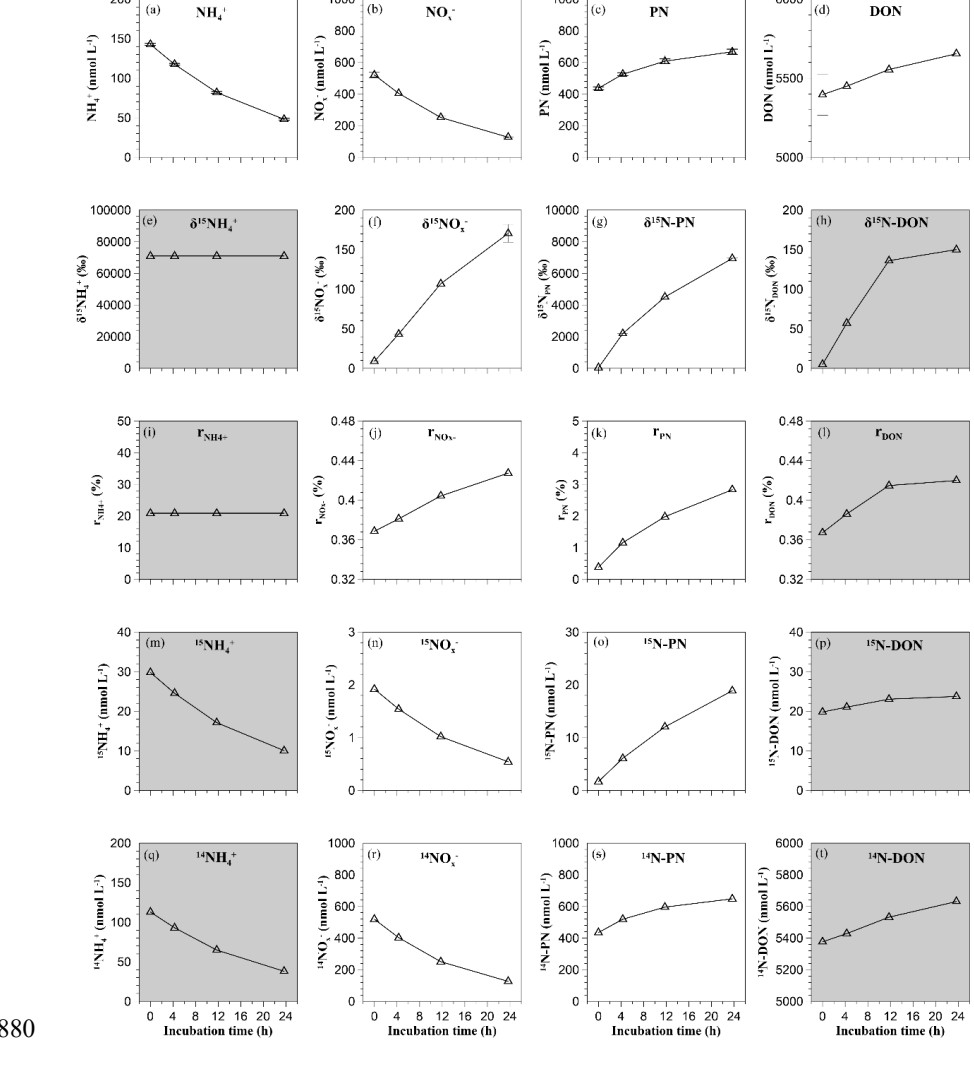



**Figure 4**





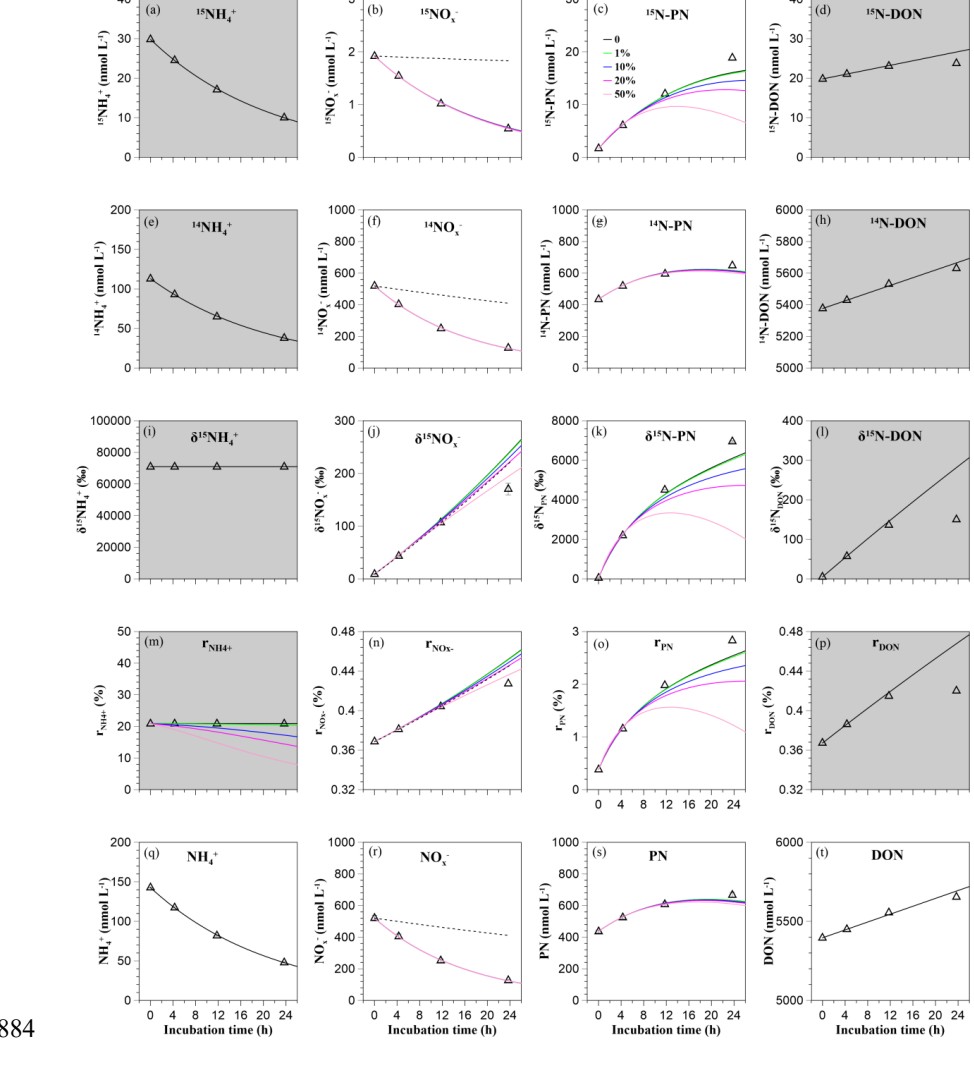

**Figure 5**





**Table 1**

| $r_{NH4+}$ | F1 | F2 | F3 | F4 | F5 | F6 | F7 | F8 |
|---|---|---|---|---|---|---|---|---|
| Decrease* | $NH_4^+$ uptake | $NH_4^+$ regeneration | $NO_3^-$ uptake | $NH_4^+$ oxidation | $NO_3^-$ uptake | $NO_2^-$ oxidation | $NO_2^-$ release | DON release |
| (%) | | | | $\mu$mol L$^{-1}$ h$^{-1}$ | | | | |
| 0 | 0.63 | 0 | 0.032 | 0.00090 | 0.22 | 0 | 0 | 0.25 |
| 1% | 0.65 | 0.014 | 0.032 | 0.00090 | 0.22 | 0 | 0 | 0.22 |
| 10% | 0.78 | 0.15 | 0.032 | 0.00090 | 0.22 | 0 | 0 | 0.22 |
| 20% | 0.94 | 0.30 | 0.032 | 0.00091 | 0.22 | 0 | 0 | 0.22 |
| 50% | 1.41 | 0.77 | 0.032 | 0.00093 | 0.22 | 0 | 0 | 0.22 |

*The $r_{NH4+}$ decrease (%) represents the total change in $r_{NH4+}$ to the end of the incubation
(14.6 h)



**Table 2**

| $r_{NH4+}$ decrease (%) | k1 NH$_4^+$ uptake | k2 NH$_4^+$ regeneration | k3 Nitrification h$^{-1}$ | k4 NO$_x^-$ uptake | k5 NO$_x^-$ release |
|---|---|---|---|---|---|
| 0 | 0.045 | 0 | 0.00050 | 0.059 | 0 |
| 1% | 0.045 | 0.00012 | 0.00050 | 0.059 | 0 |
| 10% | 0.049 | 0.0012 | 0.00050 | 0.059 | 0 |
| 20% | 0.054 | 0.0024 | 0.00051 | 0.059 | 0 |
| 50% | 0.067 | 0.0062 | 0.00052 | 0.059 | 0 |





**Table 3**

| Case | Time | Process | Matrix method (this study) | Traditional rate calculation | Santoro et al.[13] (2010)* |
|---|---|---|---|---|---|
| | (h) | | | (nmol L$^{-1}$ h$^{-1}$) | |
| WYW | 0-1.6 | NH$_4^+$ uptake | 632.2 | 413.6 | - |
| WNP | 0-4.3 | NH$_4^+$ uptake | 4.58* | 3.86 * | - |
| WYW | 0-1.6 | NH$_4^+$ oxidation | 0.90 | 0.41 | - |
| WNP | 0-4.3 | Nitrification | 0.051* | 0.046* | 0.056*# |
| WNP | 0-4.3 | NO$_x^-$ uptake | 0.059* | - | 0.01*$ |

* First-order reaction; # F value by Matlab; $ k value by Matlab.