# Peer review of "Quantification of multiple simultaneously occurring nitrogen flows in the euphotic ocean"

_Biogeosciences, 2016_

## Referee Comment (RC1) · Anonymous Referee #1 · 29 Aug 2016

The authors present a ms that utilizes sets of linear equations (as a matrix model) to describe nitrogen transformations in seawater from 25 meter depth. The authors argue that 'conventional methods' for calculating rates, including nitrification, do not consider that multiple nitrogen processes are occurring simultaneously. The authors present the model, and then illustrate 2 manipulations where enriched 15NH4+ was added to determine what nitrogen pools it ends up in. They use the program STELLA to estimate parameters of their model. They conclude that NH4 regeneration is likely an important process through isotope dilution, that their model can give differing results from the 'traditional model', and that DON is likely very important. They were able to solve for multiply occurring processes. This is an interesting ms and the community will be interested in the approach. The authors, however, need to address a number of comments to make this a more significant ms.

The recognition that there are multiple nitrogen transformations is an important one, and the coupling of the model to an enrichment assay is a strong approach. Although I appreciate what the authors are doing here, the statement that they are the first to do needs to be amended, given the recent publication of Pfister et al. in BGS. Biogeosciences, 13, 3519-3531, 2016, http://www.biogeosciences.net/13/3519/2016/ ("To our knowledge, this is the first and most convenient method designed to quantitatively and simultaneously resolve complicated nitrogen transformation rates, albeit with some uncertainties."). Thus, throughout the discussion it would be appropriate to see their model compared with the differential equation model used by Pfister et al. to model multiple nitrogen transformations. The authors also need to compare their conclusions with the above study. For example, I note that this study follows processes in seawater only, while Pfister et al. includes benthic species. It would be useful for the authors to comment on the comparisons.

The ms would benefit from more direct discussion about the comparison of the models presented in this paper and other models and approaches. Nitrogen processing rates don't seem to differ much based on methodology (Table 3), with values at least being within a similar range. I find this surprising, especially given the authors' recognition of error sources (L576) The abstract states: "comparisons with conventional labeling methods are discussed" (L28) and this is too vague. Similarly, the Conclusions could be stronger and more direct.

The ms would benefit from adhering more strongly to a clear separation of methods, results, discussion. The paragraph starting L395 is a good example where this needs to be done. It might help to shorten the ms too.

Finally, although I greatly appreciate the enrichment assay, it appears to be done once. I cannot be sure based on the description given, but it appears unreplicated and that does limit the interpretation the authors can make. Starting L314, more detail is needed including how many incubations, and whether they were replicates or uniquely treated. Having the high and low nutrient assay immediately next to each other in the methods

would also lend better comparison. As is, it is looks like these assays are unreplicated and water was collected at different depths, etc.

Specific Comments Line 50, explain what is meant by the 'inventory method' Line 57 "mainly" L105, 210 – be specific about what 'new method' means L106 is STELLA a model or a method? What is meant by "The method was also validated using the STELLA model". L150 omit 'basically' L210 what is the 'incubation system'? L243 "approximately" L416 the depth of water collection for the experiment is unclear. Here it says 25 m, while elsewhere it states .3 m. L419 the final enrichment value should be given. Methods – What were the dissolved oxygen levels? Is the assumption that this is a well-oxygenated system and loss of $^{15}N$ as gas is irrelevant? L482 'result' Though I could read eqns 5, 6, 7, they are reprinted poorly due to some 'translation' issue. L552 – Is there good evidence for light inhibition? Many studies find high rates of nitrification with normal light. The authors do not need to comment so much on inhibitors – which they did not use. Table 1 caption – explain 'different $rNH_4+$ variation" . What seems to be meant here is that the authors are manipulating the values of $rNH_4+$ to mimic the effects of isotope dilution as a consequence of regeneration. Same for Table 2 caption. Table 3 – Provide a citation for the Traditional Rate Calculation (Dugdale and Wilkerson?) and cite the equation numbers used for each.

Suppl fig 1 and 2 are STELLA figs which can be confusing without equations. I did not get much out of these figs, other than the recognition that the authors used this model structure.

Throughout the ms, the authors need to check that chemical terminology is reprinted accurately. Similarly, when subscripts are used.

---

## Referee Comment (RC2) · Anonymous Referee #2 · 2 Sep 2016

Title: Quantification of multiple simultaneously occurring nitrogen flows in the euphotic ocean Author(s): M. N. Xu, Y. Wu, L. W. Zheng, Z. Zheng, H. Zhao, E. A. Laws, and S.-J. Kao MS No.: bg-2016-298 MS Type: Research article Min Nina Xu and co-workers present an original experimental design to quantify multiple nitrogen transformation processes (rates of ammonium, nitrite and nitrate uptake, ammonia oxidation; nitrite oxidation; nitrite excretion; DON release; and potentially remineralization) by adding a single 15N-labelled ammonium substrate into a single incubation system. No inhibitors were used and special attention was given to minimize the alteration of the system by adding a limited amount of tracer. Examples of field measurements are presented and different calculation methods are discussed. The article is written in a clear and understandable manner and fits well with the scope of Biogeosciences (BG). The study is worthy of publication but the authors need to address a number of comments to

improve their manuscript (ms).

I have a concern about the method used to solve the rate law equations. Here the mass balance differential equations for determining the N-transformation rates are not integrated, neither analytically or numerically. This is rather unusual and in opposition with standard methods acclaimed for the treatment of chemical reaction kinetics. Such an approach, using rates instead of the generated profile of concentrations versus time, presents serious drawbacks, namely regarding the uncertainty on the estimated parameters (rates or rate constants). Unfortunately this point is not addressed in the ms. The authors should therefore convince the reader that their method is at least as good as conventional integration methods in terms of accuracy and precision, and this requires an uncertainty assessment (see specific comments)

The authors are not the first to propose a mass balance approach to derive multiple N-transformation rates. As far as I know, such an approach was used and discussed at least in three previous publications. 1. Elskens et al., Global Biogeochemical Cycles, vol. 19, gb4028, doi:10.1029/2004gb002332, GBC-2005 2. De Brauwere et al. Chemometrics and Intelligent Laboratory Systems 76, 163– 173, CILS-2005 3. Pfister et al., Biogeosciences, 13, 3519–3531, BGS-2016. In the GBC approach, the rate law equations are analytically integrated while in the BGS, the differential equations are solved numerically using an ODE function. Currently the use of the ODE function for solving ordinary differential equations is easy to implement (see https://cran.r-project.org/web/packages/deSolve/deSolve.pdf) and the generated profile of concentrations versus time can be fitted using least squares methods (see GBC, CILS and BGS papers). It would be appropriate to address these points in the introduction, and throughout the discussion, the authors should argue why their simplified approach can be an asset when compared to the aforementioned papers.

Also I'm not convinced that adding a single 15N-labelled ammonium tracer into the incubation system allows an accurate determination of the ammonium, nitrite and nitrate uptake rates. According to me the kinetic reactions corresponding to the matrix

expressions (Eqns 16-17) with the labelling of a single ammonium substrate is under-identified. Under this condition, the 15N-labelling of PN proceeds via the uptake of ammonium and/or via nitrification and the subsequent uptakes of nitrite and nitrate. These processes are thus not independent, and may result in a multimodal optimization problem, i.e., multiple solutions providing similar responses. The authors should address this point, especially because little information is available in the ms regarding the method used to solve Eqns. (16-17).

Specific comments Line 47 –p3: What is meant by the inventory method? Line 100 – p6: The term validation is not appropriate since the Stella model is based on the reaction kinetics outlined in Fig.1, and thereby submitted to the same underlying hypotheses than Eqns (16-17). At best we can say that the matrix solutions are consistent with a model run generating concentration versus time curves through back calculation. Lines 280/281/397/399/417. Please pay attention to the number of significant decimals when reporting data (e.g. $22.3 \pm 4.3 \mu M$ or 5376.4 nM). Line 348/354: How did the authors define 'undetectable' or 'below detection limit' in their ms? Line 420 – p23: In Fig.4 a nonlinear behavior for the concentration versus time doesn't demonstrate that the rate laws follow first order. Line 438 – p23: What is meant by 'this positive offset was compensated for by organic nitrogen utilization'. Line 518 – p27: I guess it is rather an 'accurate measurement of. . .' Line 544 – p 29: 'The uncertainty estimate for this isotope matrix method is not a simple statistical question'. Yet the authors have the means to do so. If they build rate profiles from their concentration measurements, and optimize values for $F_i$ or $k_i$ (Eqns 16- 17) using a least squares method, they will get access to the uncertainty on these parameters via the variance-covariance matrix.

---

## Referee Comment (RC3) · Anonymous Referee #3 · 15 Sep 2016

Xu and coworkers present a study of nitrogen cycling processes in two contrasting surface water environments in coastal China and the northwest Pacific Ocean. They use a 'matrix method' for calculating the competing simultaneous processes of N uptake and remineralization.

My main comments on the manuscript are: 1. The manuscript would be stronger if a wider range of environments with varying relative importances of the different processes were examined. At present, the manuscript really just addresses two incubation experiments taken from high light environments.

2. The manuscript should include a deeper discussion of the results beyond just the new method, extending to the actual ecology of the processes being examined. For example, the finding that varying the remineralization rate does not affect the nitrification rate seems significant, though potentially an artifact of the samples chosen for investigation (see #1 above).

Specific comments: There is an over emphasis on the novelty of this work being 'abandoning inhibitors', as most stable isotope labeling papers in the last decade have not used inhibitors to actually calculate rates, but rather to inform specific groups of organisms that might be contributing to a specific process. This is the case for many of the papers incorrectly cited in lines 61-63.

80% surface light intensity is a very high light intensity for trying to measure nitrification. I would suggest noting in the discussion that the contribution from nitrification to 15N uptake might be considerably different at lower (e.g. 1-10%) surface irradiance. This is somewhat alluded to in lines 381-384, but the implications could be discussed more explicitly.

Line 539: Are rates (nmol L-1 h-1) or rate constants (h-1) being compared here? Clarify language. Also, the phrase 'their nitrate uptake rate' is confusing . . .I think what is meant is 'nitrate uptake calculated using their method'

Line 562: The discussion about the relevance of this research to PNM dynamics is not warranted based on the results presented here.

Table 3: The column title 'Santoro et al.' should be clarified to say 'Rate calculation of Santoro et al' and units should be clarified for all columns (see comment above about rates versus rate constants). Table 2 has the NOx uptake rate constant (k) as 0.059 h-1, but this same value is listed as a rate (nM h-1) in Table 3.

Small errors, typos, etc.

Line 57: 'manily' should be 'mainly' Line 182: is sulfamic, not sulfanilic meant here? Line 482: 'resut' should be 'result'

---

## Author Comment (AC1) · 10 Nov 2016

**Referee #1, comment #1**

The authors present a ms that utilizes sets of linear equations (as a matrix model) to describe nitrogen transformations in seawater from 25 meter depth. The authors argue that 'conventional methods' for calculating rates, including nitrification, do not consider that multiple nitrogen processes are occurring simultaneously. The authors present the model, and then illustrate 2 manipulations where enriched $^{15}NH_4^+$ was added to determine what nitrogen pools it ends up in. They use the program STELLA to estimate parameters of their model. They conclude that $NH_4^+$ regeneration is likely an important process through isotope dilution, that their model can give differing results from the 'traditional model', and that DON is likely very important. They were able to solve for multiply occurring processes. This is an interesting ms and the community will be interested in the approach. The authors, however, need to address a number of comments to make this a more significant ms.

**Author response:**

Thanks for reviewer's appreciation of the merit of our method.

**Referee #1, comment #2**

The recognition that there are multiple nitrogen transformations is an important one, and the coupling of the model to an enrichment assay is a strong approach. Although I appreciate what the authors are doing here, the statement that they are the first to do needs to be amended, given the recent publication of Pfister et al. in BGS. Biogeosciences, 13, 3519-3531, 2016, http://www.biogeosciences.net/13/3519/2016/ ("To our knowledge, this is the first and most convenient method designed to quantitatively and simultaneously resolve complicated nitrogen transformation rates, albeit with some uncertainties."). Thus, throughout the discussion it would be appropriate to see their model compared with the differential equation model used by Pfister et al. to model multiple nitrogen transformations. The authors also need to compare their conclusions with the above study. For example, I note that this study follows processes in seawater only, while Pfister et al. includes benthic species. It would be useful for the authors to comment on the comparisons.

**Author response:**

This is a very constructive suggestion. Indeed, we missed the paper by Pfister et al. (2016) while we submitted our manuscript in July. Pfister et al. (2016) applied similar approach to resolve the N cycle processes in a tidal pool. In this version, we introduced the paper by Pfister et al. (2016) in Introduction. As suggested also by Reviewer #2, we made discussions and comparisons with their method for tidal pool.

The similarity is the coupled monitoring of changes in isotopic composition and concentration in multiple pools ($NH_4^+$, $NO_2^-$ and $NO_3^-$). However, dissimilarities include:(1) we focused on water column and all operationally defined nitrogen pools were measured, (2) the benthic biomass, which equals to particulate organic nitrogen in our case, were not measured in their tidal pools; thus, system level mass conservation cannot be made. Accordingly, their case did not allow discussions of DON release, which is an important process in water column; (3) they applied ODE to derive the mean rate constant by fitting parameter combination over the monitoring time course; by contrast, we use matrix (or linear programming in old version) to obtain rate/rate constant for the first two data points and then predict latter on changes.

We thoroughly revised our manuscript basing on three reviewers' comments. In this version, we modified our model structure slightly (see the reply for Reviewer #2) and discussed more in term of model structure and method for rate derivation. The rates derived by using ODE were also added into our tables for comparison. Meanwhile, we reorganized the manuscript in sequence from simple (low nutrient assay) to complex case (high nutrient assay). We believe that such a re-arrangement will be easier for readers to understand our method.

We attempt to resolve rates in water column, more specifically, in sun-lit ocean where intensive substrate competition occurs, thus, we modified our original statement to "This is a convenient method in euphotic zone to quantitatively and simultaneously resolve complicated nitrogen transformation rates, albeit with some uncertainties.".

**Referee #1, comment #3**

The ms would benefit from more direct discussion about the comparison of the models presented in this paper and other models and approaches. Nitrogen processing rates don't seem to differ much based on methodology (Table 3), with values at least being within a similar range. I find this surprising, especially given the authors' recognition of error sources (L576). The abstract states: "comparisons with conventional labeling methods are discussed" (L28) and this is too vague. Similarly, the Conclusions could be stronger and more direct.

**Author response:**

The significance of our method is to resolve multiple rates by adding one single tracer in one bottle for incubation. This cannot be achieved by conventional methods. As mentioned above, Pfister et al. (2016) did not include DON in their discussion according to under-identified benthic biomass.

We highlighted the significance of our method; however, we do not criticize traditional methods. For example, the traditional method for nitrification was usually conducted in the dark or deep water, thus, the consumption of substrate (ammonium) and product (nitrate) by phytoplankton was minimized presumably. In the dark, the traditional approach is ok; however, bias could be significant in the euphotic zone where phytoplankton competition appears. Meanwhile, similar rate values (less biased) can be obtained by traditional methods when one or two specific processes dominate the system.

As mentioned in Elskens et al. (2005), none model was perfect. For example, in a simple system without phytoplankton and light there will be no need to apply our approach. As mentioned in manuscript already, the traditional estimate for nitrification does not work under simulated *in situ* light in the euphotic zone since the end-product, nitrate, drops quickly due to intensive phytoplankton consumption. Such drop in end-product violates the assumption "end-product increase" in traditional method. Thus, dark incubation is required to limit phytoplankton uptake. The incubation in the dark, of course, does not represent "*in situ*" condition. The advantages of our method are (1) to explore the transformation of pathways for *in situ* condition, particularly, in euphotic ocean and at around the transition zones (e.g., nitracline and thermocline in the field) and (2) to examine responses of multiple metabolic pathways via manipulation experiments (e.g., pH, temperature and light).

Finally, the rate numbers for ammonium, nitrate uptakes and nitrification in Table 3 revealed difference. Ours values are 3-20% higher than those by traditional methods. Moreover, nitrate uptake rate by our method was ~6 times higher (in Table 3) than that derived from the equation suggested by Santoro et al. (2010) although nitrification rate was within a similar range. In this

version, we pointed out explicitly the reasons for the offset between ours and conventional methods. We enhanced comparisons among methods in Discussion part and made a stronger statement in Conclusions. We found the sentence "comparisons with conventional labeling methods are discussed" to be improper in the Abstract. We eliminated this sentence.

**Referee #1, comment #4**

The ms would benefit from adhering more strongly to a clear separation of methods, results, discussion. The paragraph starting L395 is a good example where this needs to be done. It might help to shorten the ms too.

**Author response:**

Follow this suggestion, the manuscript was reorganized. Examples were given together now in Methods from the simple to the complex case. Details for matrix solution and sensitivity test were now mentioned in Methods first and then appeared in Results. Yet, the entire length was expanded due to additional data presentation (we added 2% light incubation for comparison as requested), methodology comparison (we added results from ODE as requested; see reply to reviewer #2) and in-depth discussions.

**Referee #1, comment #5**

Finally, although I greatly appreciate the enrichment assay, it appears to be done once. I cannot be sure based on the description given, but it appears unreplicated and that does limit the interpretation the authors can make. Starting L314, more detail is needed including how many incubations, and whether they were replicates or uniquely treated. Having the high and low nutrient assay immediately next to each other in the methods would also lend better comparison. As is, it is looks like these assays are unreplicated and water was collected at different depths, etc.

**Author response:**

We do have replicates. We added descriptions for replicates in this version. In the old version, two data points were representative of replicates. We now used regular and inverse triangles, which give a clearer image of data distribution (see panels below). Moreover, we provided a new case incubated under 2% light.

Instead of discussing the biogeochemical significance of specific processes, the scope of this paper is to propose a convenient method in the field for multiple rate measures. We agree with reviewer that replicates will be helpful indeed if we attempt to probe ecosystem biogeochemistry, however, not necessarily be helpful for a new method establishment. The rate uncertainty, in fact, was largely sourced from heterogeneity of water sample and analytical errors for isotopic composition and concentration, rather than the estimator itself.

The two assays, in fact, are for two very different yet commonly seen conditions. The simple one is for oligotrophic ocean (nitrite and nitrate were pulled together in one nitrogen pool, NOx). The complex one is for estuary and coastal water where nitrite concentrations are relatively high. According to this suggestion, we illustrated both assays together in the Methods and reorganized the manuscript from the simple case to the complex case. To convince readers the applicability of our method, we presented additional data in this version to discuss these N transformation pathways under different light conditions (see panel (a) and (b) below). All the fluxes were derived from the matrix solution for the first two time points. The full time courses were generated via equation-based simulation by using Stella. We successfully and precisely predicted

the observed time courses further illustrating the performance of our method.

(a) High nutrient case – 80% sPAR

(b) High nutrient case – 2% sPAR

[Figure]

**Referee #1, specific comments:**

Line 50, explain what is meant by the 'inventory method'

**Author response:**

We followed B. B. Ward (2011). The description is now "The inventory method (monitoring substrate and/or product changes over time) is often used …".

Line 57 "mainly"

**Author response:**

Corrected.

L105, 210 – be specific about what 'new method' means

**Author response:**

We changed to "isotope matrix" method.

L106, is STELLA a model or a method? What is meant by "The method was also validated using the STELLA model".

**Author response:**

STELLA is user-friendly software for box model construction. We realized the appearance of Stella here in Introduction is improper, thus, this sentence was removed. The descriptions about STELLA will be in Methods.

L150, omit 'basically'

**Author response:**

Omitted.

L210, what is the 'incubation system'?

**Author response:**

We changed to incubation bottle.

L243, "approximately"

**Author response:**

Corrected.

L416, the depth of water collection for the experiment is unclear. Here it says 25 m, while elsewhere it states 3 m.

**Author response:**

We presented two samples collected from different locations. One was taken from the western North Pacific (low nutrient case) and the other was from a coastal bay in southern China (high nutrient case). In this version, we mentioned the two cases together in Methods to avoid confusion.

L419, the final enrichment value should be given.

**Author response:**

The description is now "…to achieve a final tracers concentration of 30 nM.."

Methods – What were the dissolved oxygen levels? Is the assumption that this is a well-oxygenated system and loss of $^{15}N$ as gas is irrelevant?

**Author response:**

Yes. We made assumptions for oxygenated water column and short term incubation. Such assumption is common and well accepted. In this version, we provided DO saturation values for all cases.

L482 'result'

**Author response:**

Corrected.

Though I could read eqns 5, 6, 7, they are reprinted poorly due to some 'translation' issue.

**Author response:**

We tried several times and even asked editorial office's help for file translation during our initial submission. The problem was due to version of software. We will work it out.

L552 – Is there good evidence for light inhibition? Many studies find high rates of nitrification with normal light.

**Author response:**

Light depresses nitrification efficiency by either direct inhibition on AMO or resource reallocation for damage recovery. Similar to previous studies, such as Merbt et al. (2012), Smith et al. (2014) and Peng et al. (2016), we found light inhibition also in coastal China seas although some recent evidences showed that some taxa of marine AOA hold genetic capabilities to reduce oxidative stress and to repair ultraviolet damage (Luo et al., 2014; Santoro et al., 2015). In the photic ocean, besides photo damage nitrifiers need to compete with phytoplankton for substrate. This is why the abundance of AOA/B increased downward in genera and also why we establish

this method to explore competing processes under in situ light condition.

According to this question and reviewer #3's suggestion, we presented additional data for the high nutrient case. For high nutrient case, actually, water from 80% sPAR and 2% sPAR were sampled for incubation. We measured fluxes for multiple pathways for different light environments, and then discussed effects of light on various processes.

The authors do not need to comment so much on inhibitors – which they did not use.

**Author response:**

We did not criticize the usefulness of inhibitor method since to block unwanted process is the only way to obtain a more accurate rate measure for specific process while using the traditional source-product method. Although inhibitor addition was not used in isotope labelling method, similar concept was applies to reduce the interference from unwanted pathway; such as nitrification rate measurement needs to be conducted in the dark to minimize ammonium and nitrate consumptions by phytoplankton. In this version, we made a clearer statement to convey proper information for inhibitor application.

Table 1 caption – explain "different $r_{NH4+}$ variation". What seems to be meant here is that the authors are manipulating the values of $r_{NH4+}$ to mimic the effects of isotope dilution as a consequence of regeneration.

**Author response:**

Yes, we did not measure isotopic compositions for $NH_4^+$. Thus, after obtaining fluxes (or rate constants) we set $r_{NH4+}$ as variable to examine the significance of remineralization in short term incubation. Results showed that remineralization would be effective in our case when incubation is prolonged over 16 hours. According to the validation by consecutive observations, remineralization is limited in all our cases in the first few hours. We added more discussions about the sensitivity test for remineralization.

We mentioned in the manuscript that once the technique for isotopic composition of low concentration $NH_4^+$ is mature (open ocean case) or in any case $r_{NH4+}$ time course was measured (coastal ocean case), all rates including remineralization can be obtained directly. Here in our case, we simulated the time courses of different nitrogen pools and assessed the importance of regeneration by manipulating $r_{NH4+}$.

Same for Table 2 caption. Table 3 – Provide a citation for the Traditional Rate Calculation (Dugdale and Wilkerson, 1986) and cite the equation numbers used for each.

**Author response:**

Reference was added.

Suppl. fig 1 and 2 are STELLA figs which can be confusing without equations. I did not get much out of these figs, other than the recognition that the authors used this model structure.

**Author response:**

We added equations into the two panels in this version.

Throughout the ms, the authors need to check that chemical terminology is reprinted accurately. Similarly, when subscripts are used.

**Author response:**

Thanks for reminding. We checked these terms carefully and will try our best to solve problems caused by format translation.

---

## Author Comment (AC2) · 10 Nov 2016

Dear Editor,

We found all comments are constructive. However some questions regarding methods were raised due to our under-descriptions in old version. We also modified slightly the model structure for both low and high nutrient cases in order to examine the processes of remineralization, DON release and ammonium uptake by microbes (< 0.7 um). As replied below, our method was simply the integration of the first two time points (trapezoid method) and unique solution can be obtained. According to comments below, we applied ODE and present the ODE-derived results in tables for comparison. The differences in rate or rate constants were caused by length for time integration. However, we need to emphasize this paper is not a model paper. The constructed box model was based on our questions just to project these full time courses of oft-measured pools for validation. Addition to the model extrapolation, we ran sensitivity test for these rate numbers to convince readers the reliability of matrix-derived values. The manuscript had been reorganized and the part of methodology was thoroughly revised.

On behalf of all the authors

Sincerely,

Shuh Ji Kao

**Reviewer #2, comment #1**

Min Nina Xu and co-workers present an original experimental design to quantify multiple nitrogen transformation processes (rates of ammonium, nitrite and nitrate uptake, ammonia oxidation; nitrite oxidation; nitrite excretion; DON release; and potentially remineralization) by adding a single [15]N-labelled ammonium substrate into a single incubation system. No inhibitors were used and special attention was given to minimize the alteration of the system by adding a limited amount of tracer. Examples of field measurements are presented and different calculation methods are discussed. The article is written in a clear and understandable manner and fits well with the scope of Biogeosciences (BG). The study is worthy of publication but the authors need to address a number of comments to improve their manuscript (ms).

**Author response:**

Thanks.

**Reviewer #2, comment #2**

I have a concern about the method used to solve the rate law equations. Here the mass balance differential equations for determining the N-transformation rates are not integrated, neither

analytically or numerically. This is rather unusual and in opposition with standard methods acclaimed for the treatment of chemical reaction kinetics. Such an approach, using rates instead of the generated profile of concentrations versus time, presents serious drawbacks, namely regarding the uncertainty on the estimated parameters (rates or rate constants). Unfortunately this point is not addressed in the ms. The authors should therefore convince the reader that their method is at least as good as conventional integration methods in terms of accuracy and precision, and this requires an uncertainty assessment (see specific comments).

**Author response:**

To avoid confusion, we change dC/dt into $\Delta C/\Delta t$.
The rate constants were determined by using the measurements at time zero and the first time point after that. The matrix equations were not constructed for calculating derivatives but to integrate the differential equation between the first two time points, and then to estimate the "instant flux" (F or k* $\overline{C}$, if time for incubation is short enough). Note that the use of "instant" here is just to make it distinguishable from the longer time incubation (or > two time points). The method we used was a second-order Runge-Kutta method, more specifically, the improved Euler method, to carry out the integration numerically. In our case, we inverted the solutions to solve for the fluxes or rate constants that would give us the correct answers at the first time point. Because the fluxes and rate constants are determined entirely from the data at time zero and the first time point, our method is equivalent to integrating the functions (trapezoid method).
After having the "instant rate" for the first time interval, we constructed a box model (equation-based input-output box model) to predict (i.e., extrapolation) the full time courses for all nitrogen pools. In previous version, the model structures and the numbers of equations and unknowns for the two cases were different (see below). However, we did not described clearly in old version. More details will be given regarding the derivation procedure. The number of equation equals the number of unknown. Thus, no uncertainty exists for the matrix solution. However, reviewer is correct for the uncertainty induced by limited data points for derivation. The major uncertainty will be sourced from analytical uncertainties and sample heterogeneity. However, in all our cases, these extrapolations agreed well with consecutive observations, suggesting a good performance of our estimator for rate or rate constant with good measurements. In this version, we stated explicitly that researchers can applied our approach by using more observational data (enlarged trapezoid) to get an average rate for longer duration if ignorable community change can be assured (see example blow for low nutrient case).
According to reviewers' comments, we modified the model slightly (see revised model structures below for comment #4) and described the two cases together in Methods.

**Reviewer #2, comment #3**

The authors are not the first to propose a mass balance approach to derive multiple N-transformation rates. As far as I know, such an approach was used and discussed at least in three previous publications. 1. Elskens et al., Global Biogeochemical Cycles, vol. 19, gb4028, doi:10.1029/2004gb002332, GBC-2005 2. De Brauwere et al. Chemometrics and Intelligent Laboratory Systems 76, 163– 173, CILS-2005 3. Pfister et al., Biogeosciences, 13, 3519–3531,

BGS-2016. In the GBC approach, the rate law equations are analytically integrated while in the BGS, the differential equations are solved numerically using an ODE function. Currently the use of the ODE function for solving ordinary differential equations is easy to implement (see https://cran.rproject.org/web/packages/deSolve/deSolve.pdf) and the generated profile of concentrations versus time can be fitted using least squares methods (see GBC, CILS and BGS papers). It would be appropriate to address these points in the introduction, and throughout the discussion, the authors should argue why their simplified approach can be an asset when compared to the aforementioned papers.

**Author response:**

Following this comment, we changed our statement about "mass balance". The statement is now "This is a convenient method specifically for euphotic zone to quantitatively and simultaneously resolve complicated nitrogen transformation rates, albeit with some uncertainties.". Above mentioned models had been referred in revision.

The rate derived from ODE is a mean of integration over time that requires a concentration time course (three points at least) for iteration and integration, thereby differs from our "instant rate" determined by two time points as replied above. Although our method is simple mathematically, we do integration. We agree with reviewer that ODE may have advantages with the support of longer time course, however, our two-point matrix solution also gave good performance for extrapolation (see figures for comment #5 by Reviewer #1).

Nevertheless, we applied ODE and made a comparison for fixed $r_{NH4+}$ condition (see the example below for high nutrient case with 80% sPAR). The rate values obtained by matrix and ODE were consistent. The difference in rate, if any, was caused by the duration for integration, i.e., shorter time (two points for the first ~2 hours) for ours and longer time (5 points for ~15 hours) for ODE. Since time series monitoring in prolonged on-deck incubation is inconvenient and inappropriate due to rapid nutrient turnover and microbial community change. Thus, we select two time point for integration. The model was constructed to reduce the potential bias in traditional source-product method caused by $^{15}N$ flows among boxes. Our aim is to provide a less biased and convenient (in term of on-deck implementation and post-hoc data processing) measure for multiple transformation rates (more specifically, the "instant rate" researchers are eager to know). As indicated by Elskens et al. (2005), over complex models can misinterpret part of the random noise as relevant processes. These boxes, i.e. PN, nitrate, nitrite and ammonium, were the most often measured species and these exchanges we applied among pools were the most fundamental processes.

**Table 2a**. Results of high nutrient case under 80% PAR.

| Rate ($k*\ \bar{C}$) nmol L$^{-1}$ h$^{-1}$ | The percentage of $r_{NH4+}$ decrease in 15 h | | | | | |
| --- | --- | --- | --- | --- | --- | --- |
| | 0 | | 1% | 10% | 20% | 50% |
| | ODE* | Isotope Matrix | Isotope Matrix | Isotope Matrix | Isotope Matrix | Isotope Matrix |
| NH$_4^+$ uptake (F1) | 361 | 397 | 397 | 399 | 401 | 408 |
| Remineralization (F2) | 0 | 0 | 21 | 211 | 424 | 1043 |
| NO$_2^-$ uptake (F3) | 28 | 29 | 29 | 29 | 29 | 29 |
| NH$_4^+$ oxidation (F4) | 1.1 | 0.4 | 0.4 | 0.4 | 0.4 | 0.4 |
| NO$_3^-$ uptake (F5) | 189 | 149 | 149 | 149 | 149 | 149 |
| NO$_2^-$ oxidation (F6) | 1.1 | 0 | 0 | 0 | 0 | 0 |
| DON release (F7) | 0 | 0 | 0 | 0 | 0 | 0 |
| Bacteria uptake NH$_4^+$ (F8) | 268 | 282 | 303 | 490 | 701 | 1314 |

*Ordinary Differential Equation

**Reviewer #2, comment #4**

Also I'm not convinced that adding a single [15]N-labelled ammonium tracer into the incubation system allows an accurate determination of the ammonium, nitrite and nitrate uptake rates. According to me the kinetic reactions corresponding to the matrix expressions (Eqns 16-17) with the labelling of a single ammonium substrate is underidentified. Under this condition, the [15]N-labelling of PN proceeds via the uptake of ammonium and/or via nitrification and the subsequent uptakes of nitrite and nitrate. These processes are thus not independent, and may result in a multimodal optimization problem, i.e., multiple solutions providing similar responses. The authors should address this point, especially because little information is available in the ms regarding the method used to solve Eqns. (16-17).

**Author response:**

Reviewer is correct about the optimization problem. We did not make clear and correct descriptions in our old version. The problem no longer exists after our modification.

In previous version for the simple case (low nutrient open ocean, include NO$_2^-$ into NO$_x^-$ as one pool), we set nitrite release from the PN pool along with $r_{PN}$ (F5 in panel (a) of Figure r1 below). We found this not reasonable since nitrite release occurs during intra-cell nitrate reduction. Meanwhile, this flux should be minor relative to other fluxes. In this version, we modified the model structure, of which the nitrite release was included as an internal cycle inside the NO$_x^-$ pool, which can be precisely measured by bacteria method. On the other hand, the remineralization input of NH$_4^+$ (F2) was connected to the DON pool instead of PN to more realistically reflect the dilution effect. As mentioned in our manuscript, once the isotopic composition of ammonium at the end-point can be measured accurately, no assumption or sensitivity test for $r_{NH4+}$ is needed. Currently, we manipulated $r_{NH4+}$ values to examine the effect of

remineralization. Via our extrapolation process, the effect of remineralization was evaluated.

According to other reviewer's suggestion, we discussed the missing nitrogen for both high and low nutrient cases (see the example of F5 and F6 in panel (b) of Figure r1 below for low nutrient case), which had been pointed out yet unresolved in previous study by Laws (1985). Here in this version, F5 in low nutrient case was the DON release from PN following the isotope ratio of PN and F6 was defined as ammonium uptake by microbes that passed through the GF/F filter (0.7 μm).

The modified model structure is shown below in (b) accompanied with the old one in (a) for comparison. According to this modification, unique solution can be obtained by matrix (6 unknowns and 6 independent equations).

[Figure]

Figure r1. The old (a) and revised (b) model structures for low nutrient case.

Equations for the low nutrient case:

$$\frac{\Delta\left[{}^{15}NH_4^+\right]}{\Delta T} = \overline{F_2} \times 0.00366 - \overline{F_1} \times \overline{r_{NH_4^+}} - \overline{F_3} \times \overline{r_{NH_4^+}} - \overline{F_6} \times \overline{r_{NH_4^+}}$$

$$\frac{\Delta\left[{}^{15}NO_x^-\right]}{\Delta T} = \overline{F_3} \times \overline{r_{NH_4^+}} - \overline{F_4} \times \overline{r_{NO_x^-}}$$

$$\frac{\Delta\left[{}^{15}PN\right]}{\Delta T} = \overline{F_1} \times \overline{r_{NH_4^+}} + \overline{F_4} \times \overline{r_{NO_x^-}} - \overline{F_5} \times \overline{r_{PN}}$$

$$\frac{\Delta\left[{}^{14}NH_4^+\right]}{\Delta T} = \overline{F_2} \times (1 - 0.00366) - \overline{F_1} \times (1 - \overline{r_{NH_4^+}}) - \overline{F_3} \times (1 - \overline{r_{NH_4^+}}) - \overline{F_6} \times (1 - \overline{r_{NH_4^+}})$$

$$\frac{\Delta\left[{}^{14}NO_x^-\right]}{\Delta T} = \overline{F_3} \times (1 - \overline{r_{NH_4^+}}) - \overline{F_4} \times (1 - \overline{r_{NO_x^-}})$$

$$\frac{\Delta\left[{}^{14}PN\right]}{\Delta T} = \overline{F_1} \times (1 - \overline{r_{NH_4^+}}) + \overline{F_4} \times (1 - \overline{r_{NO_x^-}}) - \overline{F_5} \times (1 - \overline{r_{PN}})$$

For the high nutrient complex case ($NO_2^-$ and $NO_3^-$ in separable pools), we indeed encountered equifinality problem in old version since we have 6 independent equations and 7 unknowns.

In previous version, we applied linear programming (Excel solver) to obtain the optimal solution for 7 unknowns. The non-linear GRG (Generalized Reduced Gradient Algorithm) was selected. The target function is the root mean square error for all six equations. When the value of target function reaches minimum the optimal solution was provided. After obtaining the optimal solution, we simulate time courses by using the constructed Stella model. Time course extrapolation provided by Stella were surprisingly good, thus, we overlooked the multimodel

optimization problem pointed out by reviewer.

The old and revised model are shown below in (a) and (b), respectively, for high nutrient case. Similar to the simple case, in this version we removed F7, nitrite excretion (see panel (a) below in Figure r2). In high $NH_4^+$ estuary and coastal sea, nitrate assimilation may be inhibited in oxygenated water and subsequently, the nitrite release. Thus, the ignorance of nitrite release from PN (F7 in lower panel (a)) should be acceptable. In old version, equations for PN pool were not applied independently. In order to discuss the missing ammonium, we now introduced PN into equation set. Thereby, the number of total parameters is eight. With eight independent equations a unique parameter combination can be obtained (see equations below). During the revision, we compared with ODE-derived results (see reply to comment #3 above, new Table 2a). We also examined the sensitivity of parameters in accordance with the target function (see reply below to the last specific comment) and found all rates converged to unique solutions.

[Figure]

Figure r2. The old (a) and revised (b) model structures for high nutrient case.

Equations for the high nutrient case:

$$\frac{\Delta\left[^{15}NH_4^+\right]}{\Delta T} = \overline{F_2} \times 0.00366 - \overline{F_1} \times \overline{r_{NH_4^+}} - \overline{F_4} \times \overline{r_{NH_4^+}} - \overline{F_8} \times \overline{r_{NH_4^+}}$$

$$\frac{\Delta\left[^{15}NO_2^-\right]}{\Delta T} = \overline{F_4} \times \overline{r_{NH_4^+}} - \overline{F_3} \times \overline{r_{NO_2^-}} - \overline{F_6} \times \overline{r_{NO_2^-}}$$

$$\frac{\Delta\left[^{15}NO_3^-\right]}{\Delta T} = \overline{F_6} \times \overline{r_{NO_2^-}} - \overline{F_5} \times \overline{r_{NO_3^-}}$$

$$\frac{\Delta\left[^{15}PN\right]}{\Delta T} = \overline{F_1} \times \overline{r_{NH_4^+}} + \overline{F_3} \times \overline{r_{NO_2^-}} + \overline{F_5} \times \overline{r_{NO_3^-}} - \overline{F_7} \times \overline{r_{PN}}$$

$$\frac{\Delta\left[^{14}NH_4^+\right]}{\Delta T} = \overline{F_2} \times (1-0.00366) - \overline{F_1} \times (1-\overline{r_{NH_4^+}}) - \overline{F_4} \times (1-\overline{r_{NH_4^+}}) - \overline{F_8} \times (1-\overline{r_{NH_4^+}})$$

$$\frac{\Delta\left[^{14}NO_2^-\right]}{\Delta T} = \overline{F_4} \times (1-\overline{r_{NH_4^+}}) - \overline{F_3} \times (1-\overline{r_{NO_2^-}}) - \overline{F_6} \times (1-\overline{r_{NO_2^-}})$$

$$\frac{\Delta\left[^{14}NO_3^-\right]}{\Delta T} = \overline{F_6} \times (1-\overline{r_{NO_2^-}}) - \overline{F_5} \times (1-\overline{r_{NO_3^-}})$$

$$\frac{\Delta\left[^{14}PN\right]}{\Delta T} = \overline{F_1} \times (1-\overline{r_{NH_4^+}}) + \overline{F_3} \times (1-\overline{r_{NO_2^-}}) + \overline{F_5} \times (1-\overline{r_{NO_3^-}}) - \overline{F_7} \times (1-\overline{r_{PN}})$$

Our approach differs from that in Pfister et al. (2016), in which the ratio of ammonium uptake to nitrate uptake was fixed by trial and error. According to comments below and from other reviewers, we also presented additional case and discussed the light effect.

**Specific comments**

Line 47 – p3: What is meant by the inventory method?
**Author response** – We followed B. B. Ward (2011). We made a more clear description. The sentence is now "The inventory method (monitoring substrate and/or product change over time) is often used …".

Line 100 – p6: The term validation is not appropriate since the Stella model is based on the reaction kinetics outlined in Fig.1, and thereby submitted to the same underlying hypotheses than Eqns (16-17). At best we can say that the matrix solutions are consistent with a model run generating concentration versus time curves through back calculation.
**Author response** – We partly agree with reviewer. This question was raised due to the under-descriptions of our method. For both cases, the "instant rate" for the first time interval was obtained and then served as prescribed values in Stella box model to predict the time course and to compare with consecutive observations. Since the rate is concentration dependent (first order reaction), the rate constant derived from the first time interval would not guarantee a good performance for the full time course due to decline of substrate and contemporary community change. The extrapolation is a kind of validation.
In the ocean, the rate we are eager to know is the *in situ* rate (or the instant rate at the time of sampling) before microbial community changes. Thus, short-term incubation was suggested in our previous version. Stand on this point, "validation" is a proper term.

Lines 280/281/397/399/417. Please pay attention to the number of significant decimals when reporting data (e.g. 22.3± 4.3 µM or 5376.4 nM).
**Author response** – Carefully checked and corrected.

Line 348/354: How did the authors define 'undetectable' or 'below detection limit' in their ms?
**Author response** – We change to "below the detection limit".

Line 420 – p23: In Fig.4 a nonlinear behavior for the concentration versus time doesn't demonstrate that the rate laws follow first order.
**Author response** – Reviewer is correct. Now an assumption of first order reaction was made explicitly instead of by the judgement from apparent non-linear behavior.

Line 438 – p23: What is meant by 'this positive offset was compensated for by organic nitrogen utilization'.
**Author response** – We admit the old sentence was confusing. The sentence is now "Since both ammonium and $NO_x^-$ fitted well within 12 hours, the extra non-fitted PN at the time point of 12 hours in observation likely indicated an additional nitrogen source, such as organic nitrogen, was utilized by phytoplankton when inorganic nitrogen reached threshold levels (Sunda and Hardison, 2007)." In fact, our flow cytometry data (see panel below) showed clearly the cell abundance of pico-eukaryotes increased within the first 24 hours and then decreased rapidly, very likely due to nutrient limitation. By contrast, the *Synechococcus* grew continuously even when ammonium and nitrate was around the limiting level. *Synechococcus* may thus uptake DON or recycled nitrogen

for growth. Such result is not only supportive of the importance of short-term incubation also indicative of rate might change rapidly due to community change.

[Figure]

Figure r3. The variations of cell abundances of *Pico-eukaryote* and *Synechococcus* determined by flow-cytometry.

Line 518 – p27: I guess it is rather an 'accurate measurement of…'
**Author response** – Changed as suggested.

Line 544 – p 29: 'The uncertainty estimate for this isotope matrix method is not a simple statistical question'. Yet the authors have the means to do so. If they build rate profiles from their concentration measurements, and optimize values for Fi or ki (Eqns 16- 17) using a least squares method, they will get access to the uncertainty on these parameters via the variance-covariance matrix.

**Author response** – This question was raised due to more equations for unknowns. As replied above, after revision unique solution can be obtained via the matrix method. Thus, this specific uncertainty question does not exist. We mentioned uncertainties in previous version since we also cannot deny uncertainties caused by chemical analyses. Meanwhile, errors along the time course might possibly come from the community change as replied above.

According to this suggestion, we applied a sensitivity test (see Figure r4 below) by using Excel for the low nutrient case under $r_{NH4+}$ constant condition. We set reasonable ranges for parameters and then conducted 10000 times random selection for individual parameters within the given range to generate 10000 sets of parameter combination for RMSE estimate. We can see clearly randomly selected parameters converge toward the unique solution we obtained (red inverse triangle). The RMSE is near zero. Such consistency suggests uncertainties will be sourced from chemical analyses and the heterogeneity of water for incubation rather than method itself.

[Figure]

Figure r4. The sensitivity test of parameters. Root mean square error was applied as performance measure. Inverse triangle stands for unique solution from isotope matrix method.

---

## Author Comment (AC3) · 10 Nov 2016

**Reviewer #3, Comment #1**

The manuscript would be stronger if a wider range of environments with varying relative importance of the different processes were examined. At present, the manuscript really just addresses two incubation experiments taken from high light environments.

**Author response:**

The coastal case, in fact, we sampled two layers with different light intensity, 80% and 2% sPAR, and bottles were incubated in neutral density-screened incubator to simulate original light. In the old version, we did not presented entire data since the scope of this paper is to provide a convenient method.

According to this suggestion, we presented additional data. We saw higher rates of ammonium, nitrite and nitrate uptake for the high light layer. While nitrite and ammonium oxidation were both low compared with phytoplankton uptake. The overall low ammonium oxidation rate was likely attributable to the low temperature in winter. The amount of ammonium uptake by microbes was similar to that by phytoplankton in both cases underscoring the importance of ammonium flow to < 0.7 μm particle fraction.

We do not add further cases we have. Our next paper will be focusing on the application of this method to discuss the temperature and light effect on multiple processes in an estuary along salinity gradient.

**Reviewer #3, Comment #2**

The manuscript should include a deeper discussion of the results beyond just the new method, extending to the actual ecology of the processes being examined. For example, the finding that varying the remineralization rate does not affect the nitrification rate seems significant, though potentially an artifact of the samples chosen for investigation (see #1 above).

**Author response:**

We agree with reviewer, the results might be very different in other environments. We included the layer with 2% sPAR for discussion. According to this comment, we modified the model structure (see reply to comment #4 by Reviewer #2) to discuss the missing ammonium. In old version, the unbalanced nitrogen was assigned as a leakage to DON from PN. As indicated in our manuscript, PON was operationally defined (on GF/F filter pore size of 0.7 μm). The nitrogen leakage, in fact, had been observed elsewhere. As pointed out by Laws (1985), the leakage from PON to DON or bacterial ammonium uptake (<0.7μm, absence on filter) may account for the vanishing $^{15}NH_4^+$ on PON. In this version, we separated the missing nitrogen into account for the vanishing ammonium in incubation bottle. Thus, variable remineralization rate (variable $r_{NH4+}$) was assigned to test the dilution effect.

Basing on our observational data, the continuously decreasing ammonium over time was obvious, suggesting that remineralization was insufficiently high to maintain the ammonium at steady state. Such rapid drop in ammonium was supportive of low remineralization rate deduced from

time course extrapolation. As indicated by Pfister et al. (2016), benthic mussels play a critical role in ammonium supply in tidal ponds. In our both cases, micro-zooplankton in sampled water may not present in high abundance. Limited zooplankton (animals) in sampled water is likely the key for insignificant remineralization. More discussions will be made for ecological implications.

**Specific comments**

There is an over emphasis on the novelty of this work being 'abandoning inhibitors', as most stable isotope labeling papers in the last decade have not used inhibitors to actually calculate rates, but rather to inform specific groups of organisms that might be contributing to a specific process. This is the case for many of the papers incorrectly cited in lines 61-63.

**Author response** – We do not mention 'abandoning inhibitors' in this version. References were carefully checked and cited accordingly.

80% surface light intensity is a very high light intensity for trying to measure nitrification. I would suggest noting in the discussion that the contribution from nitrification to $^{15}$N uptake might be considerably different at lower (e.g. 1-10%) surface irradiance. This is somewhat alluded to in lines 381-384, but the implications could be discussed more explicitly.

**Author response** – We totally agree with this comment. We provide low light case in this version. However, nitrification rate was still low due to low temperature in winter. We described the light effect on nitrification and referred to papers about light inhibition and substrate competition (Smith et al. 2014; Peng et al. 2016). We also explicitly stated these flows or rates in low light environment could be very different from results we presented in this study.

Line 539: Are rates (nmol L$^{-1}$ h$^{-1}$) or rate constants (h$^{-1}$) being compared here? Clarify language. Also, the phrase 'their nitrate uptake rate' is confusing . . .I think what is meant is 'nitrate uptake calculated using their method'

**Author response** – Corrected. All units in tables were carefully checked. Both rate values and rate constant will be presented clearly.

Line 562: The discussion about the relevance of this research to PNM dynamics is not warranted based on the results presented here.

**Author response** –We removed PNM relevant discussions.

Table 3: The column title 'Santoro et al.' should be clarified to say 'Rate calculation of Santoro et al' and units should be clarified for all columns (see comment above about rates versus rate constants). Table 2 has the NOx uptake rate constant (k) as 0.059 h$^{-1}$, but this same value is listed as a rate (nM h$^{-1}$) in Table 3.

**Author response** – The column title is corrected. We carefully checked for rate and rate constants throughout the manuscript and tables. The units and associated descriptions are now consistent.

Line 57: 'manily' should be 'mainly'.

**Author response** – Corrected.

Line 182: is sulfamic, not sulfanilic meant here?
**Author response** – Corrected.

Line 482: 'resut' should be 'result'.
**Author response** – Corrected.

---

## Author Response (AR1)

**To The Associate Editor**

Sub: Submission of manuscript "Quantification of multiple simultaneously occurring nitrogen flows in the euphotic ocean" for publication in *Biogeosciences*.

Dear Prof. Middelburg,

Thank you for your assistance on the process of our manuscript.

To more clearly present the capability of our method, we reorganized the manuscript and enhanced the discussion. We followed reviewer's suggestions introducing the simple (low nutrient) first and then the complex case (high nutrient). Such sequence remains throughout the manuscript.

We modified the model slightly to explore more processes associated with  $NH_{4^+}$ , including remineralization and ammonium uptake by bacteria, both were previously found important in incubation experiments. However,  $NO_2^-$  release from PON pool was removed by assuming nitrate reduction is minor among N processes. Since nitrate reduction is an intra cellular process, we also assume nitrite release would follow the r value of nitrate pool not influencing the isotopic composition of  $NO_x^-$  (nitrite and nitrate were pulled into one compartment,  $NO_x^-$ , in the low nutrient case), thus, the determination of other  $NO_x^-$  pool associated processes. Nitrite release was removed due to similar reasons for the high nutrient case. After removing one and adding two unknowns, both low and high nutrient cases can be expressed by matrix equations with unique solutions according to 14N and 15N mass balances of system level.

Many questions were raised by reviewers due to our under-description and mathematical notation regarding equations and derivation. All the equations were re-written by replacing dC/dt with  $\Delta C/\Delta t$ . We clarify all points raised and introduced details about rate derivations. We applied the first two points to calculate process rates, differing from that via ordinary differential equation (ODE, time course required). Although unique solution can be obtained, we still applied ODE for comparison (results in new tables). We did sensitivity test for the unique solution, however, the

results were shown in the reply not in the updated main text to avoid distractions. According to Reviewer#3's comments, we also added one more experimental case of low light water (2% sPAR) to reveal more ecological implications. More discussions in comparisons with previous models in terms of model structure and rate numbers were added into Discussion. The time course projection provided by STELLA was termed as validation rather than back calculation since we predict the temporal variation for 24 hours by using the rate numbers derived from the first two time points. More biogeochemical implications, such as remineralization and phytoplankton community succession, the contribution of nitrification to new production, nutrient preference and the ammonium consumption pathways, were made separately in the Section 2 of Discussion.

According to additional descriptions and discussions, the total length of this version was similar to that of the old version. However, the scientific level of this paper was promoted due to constructive comments from reviewers. We believe the current version is qualified for publication in Biogeosciences.

Yours sincerely,

Shuh-Ji Kao

高村圣

December 9, 2016

**Referee #1, comment #1**

The authors present a ms that utilizes sets of linear equations (as a matrix model) to describe nitrogen transformations in seawater from 25 meter depth. The authors argue that 'conventional methods' for calculating rates, including nitrification, do not consider that multiple nitrogen processes are occurring simultaneously. The authors present the model, and then illustrate 2 manipulations where enriched 15NH4+ was added to determine what nitrogen pools it ends up in. They use the program STELLA to estimate parameters of their model. They conclude that NH4+ regeneration is likely an important process through isotope dilution, that their model can give differing results from the 'traditional model', and that DON is likely very important. They were able to solve for multiply occurring processes. This is an interesting ms and the community will be interested in the approach. The authors, however, need to address a number of comments to make this a more significant ms.

**Author response:**

Thanks for reviewer's appreciation of the merit of our method.

**Referee #1, comment #2**

The recognition that there are multiple nitrogen transformations is an important one, and the coupling of the model to an enrichment assay is a strong approach. Although I appreciate what the authors are doing here, the statement that they are the first to do needs to be amended, given the recent publication of Pfister et al. in BGS. Biogeosciences, 13, 3519-3531, 2016, http://www.biogeosciences.net/13/3519/2016/ ("To our knowledge, this is the first and most convenient method designed to quantitatively and simultaneously resolve complicated nitrogen transformation rates, albeit with some uncertainties."). Thus, throughout the discussion it would be appropriate to see their model compared with the differential equation model used by Pfister et al. to model multiple nitrogen transformations. The authors also need to compare their conclusions with the above study. For example, I note that this study follows processes in seawater only, while Pfister et al. includes benthic species. It would be useful for the authors to comment on the comparisons.

**Author response:**

This is a very constructive suggestion. Indeed, we missed the paper by Pfister et al. (2016) while we submitted our manuscript in July. Pfister et al. (2016) applied similar approach to resolve the N cycle processes in a tidal pool. In this version, we introduced the paper by Pfister et al. (2016) in Introduction. As suggested also by Reviewer #2, we made discussions and comparisons with their method for tidal pool.

The similarity is the coupled monitoring of changes in isotopic composition and concentration in multiple pools ( $NH_4^+$ ,  $NO_2^-$  and  $NO_3^-$ ). However, dissimilarities include: (1) we focused on water column and all operationally defined nitrogen pools were measured, (2) the benthic biomass, which equals to particulate organic nitrogen

in our case, were not measured in their tidal pools; thus, system level mass conservation cannot be made. Accordingly, their case did not allow discussions of DON release, which is an important process in water column; (3) they applied ODE to derive the mean rate constant by fitting parameter combination over the monitoring time course; by contrast, we use matrix (or linear programming in old version) to obtain rate/rate constant for the first two data points and then predict latter on changes.

We thoroughly revised our manuscript basing on three reviewers' comments. In this version, we modified our model structure slightly (see the reply for Reviewer #2) and discussed more in term of model structure and method for rate derivation. The rates derived by using ODE were also added into our tables for comparison. Meanwhile, we reorganized the manuscript in sequence from simple (low nutrient assay) to complex case (high nutrient assay). We believe that such a re-arrangement will be easier for readers to understand our method.

We attempt to resolve rates in water column, more specifically, in sun-lit ocean where intensive substrate competition occurs, thus, we modified our original statement to "This is a convenient method in euphotic zone to quantitatively and simultaneously resolve complicated nitrogen transformation rates, albeit with some uncertainties.".

**Referee #1, comment #3**

The ms would benefit from more direct discussion about the comparison of the models presented in this paper and other models and approaches. Nitrogen processing rates don't seem to differ much based on methodology (Table 3), with values at least being within a similar range. I find this surprising, especially given the authors' recognition of error sources (L576). The abstract states: "comparisons with conventional labeling methods are discussed" (L28) and this is too vague. Similarly, the Conclusions could be stronger and more direct.

**Author response:**

The significance of our method is to resolve multiple rates by adding one single tracer in one bottle for incubation. This cannot be achieved by conventional methods. As mentioned above, Pfister et al. (2016) did not include DON in their discussion according to under-identified benthic biomass.

We highlighted the significance of our method; however, we do not criticize traditional methods. For example, the traditional method for nitrification was usually conducted in the dark or deep water, thus, the consumption of substrate (ammonium) and product (nitrate) by phytoplankton was minimized presumably. In the dark, the traditional approach is ok; however, bias could be significant in the euphotic zone where phytoplankton competition appears. Meanwhile, similar rate values (less biased) can be obtained by traditional methods when one or two specific processes dominate the system.

As mentioned in Elskens et al. (2005), none model was perfect. For example, in a simple system without phytoplankton and light there will be no need to apply our approach. As mentioned in manuscript already, the traditional estimate for nitrification does not work under simulated *in situ* light in the euphotic zone since the end-product, nitrate, drops quickly due to intensive phytoplankton consumption. Such drop in end-product violates the assumption "end-product increase" in traditional method. Thus, dark incubation is required to limit phytoplankton uptake. The incubation in the dark, of course, does not represent "*in situ*" condition. The advantages of our method are (1) to explore the transformation of pathways for *in situ* condition, particularly, in euphotic ocean and at around the transition zones (e.g., nitracline and thermocline in the field) and (2) to examine responses of multiple metabolic pathways via manipulation experiments (e.g., pH, temperature and light).

Finally, the rate numbers for ammonium, nitrate uptakes and nitrification in Table 3 revealed difference. Ours values are 3-20% higher than those by traditional methods. Moreover, nitrate uptake rate by our method was ~6 times higher (in Table 3) than that derived from the equation suggested by Santoro et al. (2010) although nitrification rate was within a similar range. In this version, we pointed out explicitly the reasons for the offset between ours and conventional methods. We enhanced comparisons among methods in Discussion part and made a stronger statement in Conclusions. We found the sentence "comparisons with conventional labeling methods are discussed" to be improper in the Abstract. We eliminated this sentence.

**Referee #1, comment #4**

The ms would benefit from adhering more strongly to a clear separation of methods, results, discussion. The paragraph starting L395 is a good example where this needs to be done. It might help to shorten the ms too.

**Author response:**

Follow this suggestion, the manuscript was reorganized. Examples were given together now in Methods from the simple to the complex case. Details for matrix solution and sensitivity test were now mentioned in Methods first and then appeared in Results. Yet, the entire length was expanded due to additional data presentation (we added 2% light incubation for comparison as requested), methodology comparison (we added results from ODE as requested; see reply to reviewer #2) and in-depth discussions.

**Referee #1, comment #5**

Finally, although I greatly appreciate the enrichment assay, it appears to be done once. I cannot be sure based on the description given, but it appears unreplicated and that does limit the interpretation the authors can make. Starting L314, more detail is needed including how many incubations, and whether they were replicates or uniquely treated. Having the high and low nutrient assay immediately next to each other in the methods would also lend better comparison. As is, it is looks like these

assays are unreplicated and water was collected at different depths, etc.

**Author response:**

We do have replicates. We added descriptions for replicates in this version. In the old version, two data points were representative of replicates. We now used regular and inverse triangles, which give a clearer image of data distribution (see panels below). Moreover, we provided a new case incubated under 2% light.

Instead of discussing the biogeochemical significance of specific processes, the scope of this paper is to propose a convenient method in the field for multiple rate measures. We agree with reviewer that replicates will be helpful indeed if we attempt to probe ecosystem biogeochemistry, however, not necessarily be helpful for a new method establishment. The rate uncertainty, in fact, was largely sourced from heterogeneity of water sample and analytical errors for isotopic composition and concentration, rather than the estimator itself.

The two assays, in fact, are for two very different yet commonly seen conditions. The simple one is for oligotrophic ocean (nitrite and nitrate were pulled together in one nitrogen pool,  $NO_x^{-}$ ). The complex one is for estuary and coastal water where nitrite concentrations are relatively high. According to this suggestion, we illustrated both assays together in the Methods and reorganized the manuscript from the simple case to the complex case. To convince readers the applicability of our method, we presented additional data in this version to discuss these N transformation pathways under different light conditions (see panel (a) and (b) below). All the fluxes were derived from the matrix solution for the first two time points. The full time courses were generated via equation-based simulation by using Stella. We successfully and precisely predicted the observed time courses further illustrating the performance of our method.

(a) High nutrient case -80% sPAR

(b) High nutrient case -2% sPAR

---

## Author Response (AR3)

The authors have done much to address the comments of the reviewers and the revised ms is much stronger as a result. I think that readers will benefit from seeing the exchange among authors and reveiwers.

Although the ms is clearly written, the ms still needs a close editorial check for good grammar throughout.

e.g. line 99, "needful" should be "needed". L108 "Thank for recent advances . . ." There are many examples of grammatical errors that need to be corrected. I appreciate that English is not their first language, so they need the aid of a good editor or colleague to correct mistakes.

The terminology "tidal pools" should be used instead of 'ponds'.

Edward Laws, has carefully checked and thoroughly edited the manuscript in terms of language (changes are in blue in the trackable version).

Minor comments e.g. line 99, "needful" should be "needed".

Corrected.

L108 "Thank for recent advances . . ."

Follow Reviewer #3, we changed it to "As a result of".

The terminology "tidal pools" should be used instead of 'ponds'.

Changed.

**Anonymous Referee #3**

> The authors have satisfactorily responded to my previous comments, though the editing has muddled the explanation of the model somewhat. I have only minor comments at this point.

Thanks for the reviewer's recognition.

> line 108: this should be 'Thanks to' not 'Thanks for'. Probably better to substitute "As a result of"

As suggested, we changed it to "As a result of".

> The development of the model was laid out more clearly in the previous version. I am not sure the reviewer comments that brought about this change? At the very least, explicit definition of the source of the atmospheric atm% constant term (0.00366) should be given for unfamiliar readers.

Yes, additional part for model development was added according to previous review. We add the reference of Coplen et al. (1992) for the source of the atmospheric atm% constant term (0.00366).

> lines 333-334: N/P ratio below 16 doesn't necessarily mean the system is N limited. 0.5 uM [NO3-] concentration suggests it is not.

We would like to provide more background information originally to show readers P is not limiting during entire incubation. To avoid distraction, we deleted this sentence. Such deletion would not influence the logic flow and story.

> lines 392-394: This sentence is unclear.

The sentence "In general, the rates of the first time interval can well predict the following up observations, demonstrating a good predictive performance by using the matrix method instant rate." was changed to "The fact that the rates during the first time interval generally predicted rather well the subsequent observations demonstrated a good predictive performance with the matrix method initial rate."

> line 535: These are rate constants, not rates.

Corrected.

> line 569: This comment about the author's reply to reviews should not be in the final manuscript.

We decided to add this figure into supplementary information (see Fig. S3).

> line 589: I would change this to 'may have been overestimated'

Changed as suggested.

line 608-617: Double check this paragraph for correct usage of RPI, as it there are several typos where it says PRI instead.

We changed to RPI.

The verb tense for many of the sub-headings in Section 4 should be changed, e.g. 4.2.4 should say 'Quantifying' not 'Quantify'

Changed as suggested.

[revised manuscript text omitted]

$\delta^{15}$N-NO$_2^-$, (r) $\delta^{15}$N-NO$_3^-$, (s) $\delta^{15}$N-PN, (t) $\delta^{15}$N-DON, (u) [NH$_4^+$], (v) [NO$_2^-$], (w)

[NO$_3^-$] (x) [PN] and (y) [DON]. The black regular and inverse open triangles represent the duplicate observational values; the black, green, blue, magenta and pink solid lines represent the STELLA model simulations of $r_{NH4+}$ decreases 0%, 1%, 10%,

20% and 50% in 15 h, respectively.

**Fig. 1**

[Figure]

**Fig. 2**

[Figure]

**Fig. 3**

[Figure]

**Fig. 4**

[Figure]

**Fig. 5(A)**

[Figure]

[Figure]